# The Fragility of Fairness:
# Causal Sensitivity Analysis for Fair Machine Learning

**Jake Fawkes**[*]
Department of Statistics
University of Oxford
jake.fawkes@st-hughs.ox.ac.uk

**Nic Fishman**[*]
Department of Statistics
Harvard University
njwfish@gmail.com

**Mel Andrews**
Philosophy Department
University of Cincinnati

**Zachary C. Lipton**
Machine Learning Department
Carnegie Mellon University

## Abstract

Fairness metrics are a core tool in the fair machine learning literature (FairML), used to determine that ML models are, in some sense, "fair." Real-world data, however, are typically plagued by various measurement biases and other violated assumptions, which can render fairness assessments meaningless. We adapt tools from causal sensitivity analysis to the FairML context, providing a general framework which (1) accommodates effectively any combination of fairness metric and bias that can be posed in the "oblivious setting"; (2) allows researchers to investigate combinations of biases, resulting in non-linear sensitivity; and (3) enables flexible encoding of domain-specific constraints and assumptions. Employing this framework, we analyze the sensitivity of the most common parity metrics under 3 varieties of classifier across 14 canonical fairness datasets. Our analysis reveals the striking fragility of fairness assessments to even minor dataset biases. We show that causal sensitivity analysis provides a powerful and necessary toolkit for gauging the informativeness of parity metric evaluations. Our repository is available here.

## 1 Introduction

Fair machine learning (FairML) is a theoretical approach to studying and remediating disparities in prediction and allocation systems based on machine learning algorithms. A core focus of the field has been to develop, evaluate, and train models to satisfy a number of "fairness metrics". These metrics operationalize the social ideal of fairness as a statistical quantification of some performance measure compared across demographic groups. Such evaluations often play an important role in auditing ML systems [11, 51, 48] to certify whether models satisfy some tolerable level of statistical disparity.

Real-world data, however, is frequently plagued by a variety of measurement biases and other violated assumptions which can undermine the validity of fairness metrics[5, 34]. While such biases come in many forms, in this work we focus on the following: noisy or poorly-defined outcome measures (proxy bias) [27], the observation of samples or outcomes from only a subset of the population (selection bias) [6, 35], or causal impacts on outcomes through background policies within a firm's control [16], which we term *extra-classificatory policies*, or ECPs. We focus on these varieties of bias due to their ubiquity in FairML applications, which we demonstrate through an analysis of their prevalence and magnitude in a range of benchmark datasets (see Table 1 and App. D).

Motivated by the problems posed by such measurement biases, we offer a framework based on graphical causal inference to operationalize assumptions about data quality issues, alongside methods adapted from causal sensitivity analysis for statistical quantification of their impacts on fairness evaluations. This framework enables both ML practitioners and auditors to empirically gauge the

sensitivity of parity metrics to assumption violations for specific combinations of metrics, datasets, and use cases. Causal inference is particularly apt for this problem, as it provides a formal language within which to precisely identify the goals of a particular study: what is the quantity we seek to estimate, and in which population? This accounts for the success of causal inference in the social sciences and makes it similarly well-suited for use in the arsenal of ML auditing tools.

We leverage recent developments in automated discrete causal inference, particularly the autobounds framework of Duarte et al. [21], to provide a unified causal sensitivity analysis framework for the "oblivious" setting, as laid out in Hardt et al. [32]. In this setting, we only have access to protected attributes $A$, the true target labels $Y$, and the predicted labels $\hat{Y}$, but not to covariates $X$. For example, in evaluating racial discrimination in loan granting, one has access to the race attribute $A$, the true repayment rate $Y$, and the predictions $\hat{Y}$, but not to input features $X$ nor the form of $\hat{Y}(X)$.

This lends us a straightforward procedure for performing sensitivity analyses for any combination of measurement bias and suitably well-behaved metric that can be posed obliviously: (i) Express the bias in terms of a causal graph—a directed acyclic graph, henceforth DAG, (ii) choose a sensitivity parameter to control the degree of bias, (iii) provide any additional probabilistic assumptions or relevant structural knowledge. The problem of bounding a statistic under a given degree of bias can then be converted to solving a given constrained optimization problem [21], which is achieved via a branch and bound solver [8], leading to valid bounds even when a global optimum is not reached.

We apply this framework to systematically explore the sensitivity of different metrics to the three biases—proxy label bias, selection bias, and extra-classificatory policy bias—for different datasets and classifiers. Our results reveal that many fairness metrics are in fact *fragile*: realistic violations of core underlying assumptions can imply vacuously wide sensitivity bounds. In other words, features of typical deployment contexts can easily render fairness evaluations useless or uninformative.

The fragility of well-known fairness metrics to pervasive biases represents one key empirical finding. Our second core result demonstrates the existence of tradeoffs between the complexity and fragility of fairness metrics. The robustness of parity notions scales inversely with their dependence on predictive outcomes and the intricacy of this dependence function. We find demographic parity to be most robust to measurement biases, while predictive parity metrics exhibit the most fragility to bias. In light of known incommensurability results [39], we urge the importance of understanding these tradeoffs and their practical implications, for both practitioners and auditors alike.

With these experiments, we aim to demonstrate that the biases we describe are an unavoidable aspect of the FairML problem, not a mere addendum. As such, we have hopes that our unified sensitivity analysis framework can enable both auditors and practitioners to understand how robust their "fairness" evaluations are to various measurement biases by precisely articulating the quantity they wish to evaluate and its divergence from what has been measured. Finally, we hope this work will inspire greater emphasis going forward on the realities of real-world deployment scenarios, such as measurement biases, and their impacts on fairness evaluations.

## 2   Related work

**Proxy Label Bias**   Proxy Label bias is a foundational problem in FairML, endemic within criminal justice and legal applications of ML, which the fairness literature originally arose to address [5, 28]. Fogliato et al. [27] presents one of the earliest considerations of sensitivity analysis within FairML, algebraically deriving sensitivity bounds for proxy label bias. As we demonstrate in 6, our approach is capable of re-deriving and extending these results. In a similar vein, Adebayo et al. [1] studied the effects of label noise on fairness metric evaluation, although this work was largely empirical. Guerdan et al. [31] propose several causal models for reasoning about proxy labels in human—algorithm joint decision making, which can be rendered compatible with our sensitivity analysis framework. Further work has explored alternate aspects of fairness evaluations under proxy label bias [63, 64].

**Selection Bias**   Selection bias was first considered in FairML as "selective labels" by Lakkaraju et al. [43], focused on the scenario in which a policy determines which outcomes are observed. Kallus and Zhou [35] study the effects such a biased policy can have on equalized odds for the unselected population when predictors are trained only on the selected population. Various works now link selection bias in fairness to causal inference [26, 59] with Goel et al. [30] providing a summary of the different types of selection in terms of causal graphs. Coston et al. [17] take an importance-weighting approach to train FairML classifiers with selective labels, under assumptions on the structure of the

missingness. Zhang and Long [67] study how to assess the accuracy parity in unselected data, from the selected data, under assumptions on selection structure and the FairML model class.

**Extra Classificatory Policy Bias**   We investigate the impacts of extra classificatory policies, that is, policies under the control of the predicting agent which causally affect the outcome of interest. This work is related to counterfactual risk assessments [16], and subsequent work on counterfactual equalized odds [47]. Sensitivity analysis approaches have further been developed for unmeasured confounding [12, 52]. Our methodology differs from these works in our focus on the oblivious setting. As such, we focus less on identification, but rather on how influential a policy would need to be before it could significantly impact a fairness evaluation.

**Additional Related Work**   Beyond the above work on sensitivity analysis there are other general approaches to understanding the robustness of algorithmic fairness to data bias [45], such as adversarial robustness [13, 41] and distributional robustness [60]. These are very flexible in terms of the types of bias they can represent, however, this renders them less interpretable and less able to incorporate additional assumptions. One measurement bias we did not consider is proxy attribute bias [14], for which there exist quite comprehensive sensitivity analysis results [36]. There is also a selection of work performing sensitivity analysis for unmeasured confounding in causal fairness [38, 58, 65]

## 3   Measurement Biases

This section introduces the measurement biases we consider via two reccurring examples, demonstrating how they arise in the wild and emphasizing the role of practitioner choice. We first deliver a conceptual illustration of said biases (3.1-3.3) before presenting an empirical analysis of their prevalence across a range of FairML benchmark datasets [44].

### 3.1   Proxy Label Bias

We first discuss proxy label bias, introducing it via the following example:

> **An Algorithmic Hiring System:** *Suppose that a company receives thousands of applications for every job they advertise. To handle this, they elect to build an ML-based system to assist in sifting through candidates. They opt to construct a model for predicting employees' performance review scores from their resumés, which is then used to assign scores to applicants based on predicted performance reviews. Finally, to ensure that the model is fair, they check against standard fairness metrics on a held-out subset of the training data.*

The company described has taken steps to ensure that its job candidate filtering model is "fair." However, it has only run parity tests relative to the variable chosen as its target of prediction. The latent variable of interest in this scenario, what the firm ideally strives to predict, is "employee quality." However, "employee quality" is multifaceted and socially constructed; it is not a phenomenon that can be directly and objectively measured.[1] Instead, engineers leverage a *proxy label*. Unlike the nebulous property of employee quality, employee performance reviews are readily available. It is facially not unreasonable to take performance reviews to stand in for employee quality; after all, the one exists, at least putatively, to track the other. However, there is a problem with this strategy: It has been extensively documented that performance reviews are often discriminatory—in other words, performance reviews are both a worse signal of the true underlying latent for certain demographics, and are skewed negative for those demographics relative to the true latent [20, 55].

If an outcome is biased, then classifiers optimizing predictive accuracy on a proxy can appear to satisfy a fairness metric when, in reality, the metric has inherited the biases of the outcome. Practitioners must understand whether a particular outcome is well-suited to the underlying decision problem, alongside any skew or measurement bias that may be induced by using such a measure. Often the use of a proxy outcome is unavoidable. If so, sensitivity analysis is a key tool for understanding what impact biases in the proxy could have on parity metric evaluations.

### 3.2   Selection Bias:

Introducing our second example to discuss selection bias:

---

[1] For discussion of how mismatches between unobservable latents and their proxies play in fair ML, see [34].

> **An Automated Loan-Approval Algorithm:** *A large bank has an online lending platform that receives thousands of loan applications per day, the vast majority of them for under USD $1,000. The bank cannot afford to assign employees to assess all the applications, and elects to automate the process. The bank uses its data on loan repayment to fit a model for likelihood of repayment, with loans automatically approved if the estimated probability of repayment sits above some threshold. The model is once again assessed via standard fairness metrics on holdout data.*

As in the previous example, the bank only possesses repayment information for the population that has historically been approved for loans; the subset of the broader population that was not granted a loan is therefore unobserved. Each firm trains a classifier on the subpopulation for which it possesses outcome data. The classifiers' performance on the entire population of job applicants and loan applicants, respectively, are unknown. Further, fairness guarantees on only selected populations can fail to meaningfully extrapolate to deployed models, especially when historical selection procedures encode biases. This phenomenon has been referred to as *selective labels* [43] or *prejudiced data* [35], and now more commonly under the catch-all term of *selection bias* [30].

While selection bias covers the selective labels case, it encompasses a broader class of examples. An important question in the evaluation of fairness metrics is what population the practitioner would ideally want to evaluate the statistic in. We refer to this as the *reference population*. At bare minimum, the reference population should be the population the model is deployed on, not the training population—as the selective labels literature points out. However, there are situations where arguments could be made for broader reference populations. For example, should fairness metrics for an algorithmic hiring system be assessed on the local pool of applicants to that company, or the global pool of applicants to similar roles? Dai et al. [18] demonstrate that, with reputational dynamics in play, applicants may strategically apply to firms based on their chances of success. A firm's hiring practices can therefore satisfy fairness desiderata relative to their applicant pool having radically skewed the demographics of that population via a history of discriminatory hiring policies. Put crudely: the firm that "women candidates know not to apply to" is not non-discriminatory, but an evaluation of hiree demographics relative to the applicant pool might deceptively depict it as such.

There are no universalisable solutions to such issues, as the choice of reference population must depend on precisely what task the model is being trained to perform, and where it will ultimately be deployed. This further points to a need for practitioners to be clear about what population they have chosen, how it may differ from the data they have measured, and why this specific reference population is preferable to others for the task at hand.

### 3.3 Extra-Classificatory Policy Bias

A third issue emerges when considering additional context in the loan-approval case study:

> **Automated Loan-Approval (cont):** *Our bank now has a model for loan approval. However, they must also set interest rates for approved loans. For this, they employ a legacy model, which works off of a number of factors including credit score, loan amount, lendee address, and various macroeconomic variables.*

While the bank has determined that the classifier is fair in the sense that it does not discriminate according to known demographic features and relative to repayment history, these notions of fairness fail to account for the interest rate that the bank determines. As this has a direct and powerful impact on a lendee's likelihood of repayment, if it is set in an (intentionally or unintentionally) discriminatory manner, it can have a large and unaccounted-for effect on the evaluation of fairness metrics.

This is not specific to the loan approval setting; in many scenarios, firms control a number of "levers" that causally impact outcomes for classified populations—we call these *extra-classificatory policies*. Such extra-classificatory policies can drastically affect outcomes, and so shape the data on which models are trained. Through such policies, a firm can create the appearance of demographic base rate discrepancies which then, via predictive models trained on historical data, serve to justify differential lending policies across demographic subpopulations.

The issues pointed towards by extra-classificatory policies are multi-faceted and context-specific. As such, a key challenge to the FairML community is detailing such issues and forming mathematical models of them. We focus on binary policies (e.g. [16]), leaving more general settings to future work.

| Proxy Bias | Selection Bias | ECP Bias | One Bias | Two Biases | Three Biases |
|---|---|---|---|---|---|
| 69% | 85% | 85% | 100% | 92% | 61% |

Table 1: We document the proportion of realistic FairML benchmark datasets (from Le Quy et al. [44]) that exhibit each of the three biases we discuss. For more details see App. D.

## 3.4 Cross-Dataset Analysis

To demonstrate the prevalence of these issues, we analyze the presence of each measurement bias for the FairML benchmark datasets given in Le Quy et al. [44]. We remove any datasets not associated with a concrete decision problem, leaving 14 datasets from a variety of different domains, including financial risk, criminal justice, and employment/university admissions. We summarise the results in Table 1, with the full table of datasets and biases along with rationales available in Appendix D. Our results demonstrate the scope of the problem created by measurement bias, with all the datasets displaying at least one of the problems and 60% displaying all three simultaneously. From this, we see that such biases are themselves a key part of the FairML problem, not optional complications. Moreover, this points to the need for methodological solutions for evaluating and training FairML models in settings where multiple biases are present simultaneously.

## 4 Graphical Causal Sensitivity Analysis

Here we outline some technical background on graphical causal sensitivity analysis, from graphical sensitivity analysis to the autobounds framework [21] which automates discrete sensitivity analysis. In the following section, we apply this to FairML to construct a sensitivity analysis tool for oblivious settings [32], where we do not observe covariates and all other variables are discrete.

**Notation**   We let $Y$ denote the outcome the practitioner wishes to measure, $X$ the observed covariates, $A$ the protected/sensitive attribute, and $\hat{Y}$ the prediction of $Y$ with domains $\mathcal{Y}, \mathcal{X}, \mathcal{A}, \hat{\mathcal{Y}}$.

## 4.1 Causal Background

We begin by defining the structural causal model (SCM) approach to causality [49, 54]. Here we model causal relationships via deterministic functions of the observed variables and additional latent variables, with the latter representing the unobserved or random parts of the system. We will always label the observed variables $\mathbf{V}$ and the unobserved variables $\mathbf{U}$. These equations lead to a causal graph that has a node for each variable and a directed edge $V_1 \to V_2$ if $V_1$ is an argument of the function determining the value of $V_2$. To illustrate this, the following figure demonstrates the SCM and corresponding graph we assume throughout for the relationships between $A, X, Y,$ and $\hat{Y}$:

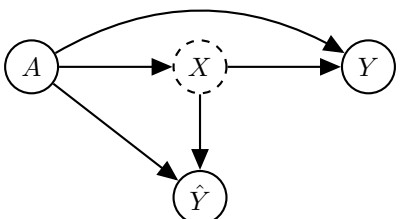

$$X = f_X(A, U_1)$$
$$Y = f_Y(X, A, U_2)$$
$$\hat{Y} = f_{\hat{Y}}(X, A, U_3)$$
$$\mathcal{C} = \left(\{f_X, f_Y, f_{\hat{Y}}\}, P(\mathbf{U})\right)$$

(a) The directed acyclic graph representing the relationships between $A, Y, \hat{Y}$ and the unobserved $X$. The latent variables are implicit in the DAG.

(b) The SCM $\mathcal{C}$ over the DAG in (a) defined by a set of functions on $\mathbf{V} = \{A, X, Y, \hat{Y}\}$ and a probability distribution over the implicit latent variables.

Figure 1: Example of a DAG and the corresponding SCM. Unobserved variables are dashed.

We will use $\mathcal{C}$ to depict an SCM, which consists of a collection of functions and a probability distribution over the noise terms. A complete definition of SCMs can be found in Appendix B.1.

**Marginalisation In Causal Models**   Often—and especially in FairML applications—we do not observe all relevant variables. For example, in this work, we assume the covariates $X$ are unobservable. However, this is less of a problem than it originally appears due to latent projection, as introduced by Verma and Pearl [62]. Latent projection allows us to marginalize out any unobserved variables

while preserving the causal structure over observed variables, as demonstrated by Evans [23]. We visualize this process in Fig. 2, where we marginalize over $X$ leaving just $A, \hat{Y},$ and $Y$ and latent variable, $U$. We outline this procedure in detail in Appendix B.2. The important point is that we can always preserve the causal structure over our observable variables by using a finite number of latent variables. This is the case regardless of how many variables we marginalize out.

## 4.2 Partial Identification and Sensitivity Analysis

We now show how structural causal models can be used to perform sensitivity analyses for causal (or non-causal) queries, first introducing the important concept of partial identification.

**Partial Identification**    In partial identification, the goal is to understand what values a particular statistic can take relative to our assumptions. We call this a *Query* and write it as a function $\mathcal{Q}(\mathcal{C})$ which takes an SCM and returns a real number. For example, if we wanted to measure counterfactual fairness (for binary $A$), we could define the query $\mathcal{Q}_{\mathrm{CF}}$ as:

$$\mathcal{Q}_{\mathrm{CF}}(\mathcal{C}) := P_{\mathcal{C}}(\hat{Y}(A = 1) \neq \hat{Y}(A = 0))$$

Where $\mathcal{Q}_{\mathrm{CF}}(\mathcal{C}) = 0$ exactly when the predictor is counterfactually fair according $\mathcal{C}$ [25].

Given a query of interest, $\mathcal{Q}$, the goal of partial identification is to understand what possible values $\mathcal{Q}$ can take given the practitioner's prior knowledge and assumptions. Here we will consider partial identification for a fixed DAG and a set of constraints on the probability distributions and functions defining the SCM. Practitioners can encode these assumptions by defining a set of SCM models, which we will write $\mathcal{M}$. The most natural example here is to let $\mathcal{M}$ contain all possible causal models arising from the graph with the same observational distribution as the measured dataset. Practitioners can restrict this set of SCMs to incorporate more domain-specific information, making the bounds more informative. Using this notation, partial identification is rendered as a pair of optimization problems that lower and upper bound the query of interest over $\mathcal{M}$:

$$\min_{\tilde{\mathcal{C}} \in \mathcal{M}} \mathcal{Q}(\tilde{\mathcal{C}}) \leq \mathcal{Q}(\mathcal{C}) \leq \max_{\tilde{\mathcal{C}} \in \mathcal{M}} \mathcal{Q}(\tilde{\mathcal{C}}) \tag{1}$$

**Sensitivity Analysis**    In sensitivity analysis, the goal is to understand how violations of assumptions affect the measure of a statistic. The initial step is to define some sensitivity parameter, which measures the degree to which an assumption is violated. For example, in the proxy attribute literature [14, 36], the goal is to understand how sensitive fairness metrics are to mismeasured protected attributes. For example, as we will discuss later in terms of proxy bias, a natural sensitivity parameter would be $P(Y_P \neq Y)$, where $Y_P$ is the proxy outcome. We may then let $\mathcal{M}_{\mathrm{Prox}}(\delta)$ be the set of causal models that have $P(Y_P \neq Y) \leq \delta$ and comply with the practitioner's assumptions. By repeatedly solving the partial identification problem for different $\delta$ to understand how large $\delta$ must be, the statistic $\mathcal{Q}$ becomes uninformative.

## 4.3 Discrete Causal Sensitivity Analysis

To perform the partial identification required for causal sensitivity analysis, we have to solve the max/min problem in (1). This problem as formulated generically is not tractably solvable, but additional structure on either the query, $\mathcal{Q}$, or set of models, $\mathcal{M}$, can lead to tractable optimization problems and computable bounds. We focus on settings where all variables are discrete, which is particularly helpful for partial identification problems [29, 9, 53] due to the function response framework [4]. This framework takes advantage of the fact that, given a causal graph $\mathcal{G}$, if we fix the latent variables $\mathbf{U}$, the structural equations are deterministic functions of their other inputs in $\mathbf{V}$. If the observed variables are discrete, there are only finitely many such functions. As a result of this, we can represent every single SCM using the one set of fixed structural equations and a distribution over some discrete latent variables, $\tilde{\mathbf{U}}$. This means that any SCM with discrete observed variables and a fixed graph $\mathcal{G}$ can be represented entirely by the distribution $P(\tilde{\mathbf{U}})$, and so by a point in the probability simplex $\Delta^k$ for some $k$ [21, 24].

Duarte et al. [21] showed that this allows partial identification problems (1) to be converted into tractable optimization problems, where now the set of causal models $\mathcal{M}$ corresponds to a subset of the probability simplex $\mathcal{M}^{\Delta} \subset \Delta^k$ and the query $\mathcal{Q}$ becomes a function $f_{\mathcal{Q}} : \Delta^k \to \mathbb{R}$. Moreover, any statement that can be written as a polynomial in probabilities over factual and counterfactual statements corresponds to a fractional polynomial in $\mathbf{p}_{\mathcal{C}}$. So, if $f_{\mathcal{Q}}$ is a polynomial in such probabilities

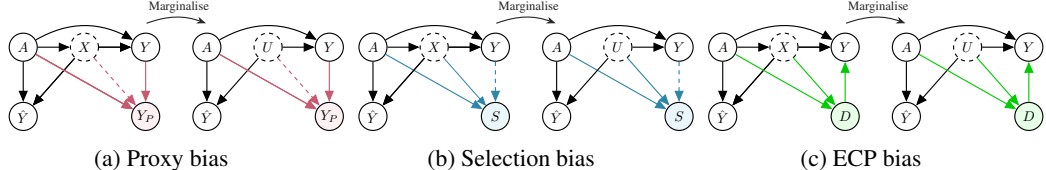

Figure 2: Causal graphs for each of the biases showing the assumed causal structure over all variables, and the implied structure upon marginalizing out $X$. Dashed lines denote varying assumptions.

and any prior assumptions can be stated via probabilistic statements and arithmetic operations, the partial identification problem can be converted into a polynomial programming problem. Duarte et al. [21] propose to solve these partial identification problems via branch and bound solvers [8], which ensures that the program produces valid bounds even if convergence is not reached.

# 5   Causal Sensitivity Analysis for FairML

We now apply the methodology outlined in Section 4.3 to the FairML setting to create a sensitivity analysis tool for parity metric evaluations. We focus on settings where $(A, Y, \hat{Y})$ are discrete and the auditor does not have access to the covariates $X$. The following steps lead to a sensitivity analysis tool for any measurement bias that can be stated obliviously and any statistic that can be written as a polynomial in factual and counterfactual probabilities:

1. Determine how the sampled population differs from the target population, expressing the difference in terms of a causal graph over all variables. Marginalize out $X$, to leave a causal structure over $(A, Y, \hat{Y})$ and any bias-specific variables.

2. Choose a sensitivity parameter to control the degree of measurement bias and provide any additional knowledge relevant to the task at hand.

3. With all this perform the sensitivity analysis by repeatedly solving the optimization problem in (1) for the test statistic using the methodology outlined in 4.3.

We apply this procedure to each of the measurement biases given in Section 3, using the causal graphs in Fig. 2 to depict the biases. Causal graphs are context-specific, so we do not expect these to be appropriate in all cases. Instead, we use them as plausible graphs for showcasing our framework.

**Proxy Label Bias**   **(1)** We represent the difference between the measured and target populations using an additional variable $Y_P$. This denotes the measured proxy of the outcome. We assume that this outcome is a noisy version of the true outcome $Y$, where the noise depends on $A$ and can optionally depend on $X$. As we show in Appendix B.3, assuming the proxy depends on additional unobservables leads to the same graph over $(Y, A, \hat{Y})$. **(2)** For the sensitivity parameter, we use the probability that the proxy differs from the outcome the practitioner hopes to measure, so $P(Y_P \neq Y)$.

**Selection Bias**   **(1)** We signal whether or not an individual is selected with a binary variable $S$, which we assume depends on an individual's protected attribute, covariates, and, in some instances, the outcome. **(2)** For the sensitivity parameter, we choose $P(S = 0)$, the probability of a sample not being selected. This controls the proportion of the population which remains unobserved. However, there are other natural choices for sensitivity parameters, such as statistical measures of sample quality [46]. For selection bias, the practitioner could have significant information about the unselected population which could be used to tighten bounds. For example, selective labels would lead to information on the covariates for the unselected populations, and thus the $(A, \hat{Y})$ proportions.

**Extra-Classificatory Policies**   There are a plurality of plausible ways to mathematically represent the problems arising from ECPs, with different formulations being better suited to different concerns. Here we proceed as follows: **(1)** We use an additional variable, $D$, to depict the policies in the firm's control, which we take to be binary. We then use counterfactual versions of parity metrics, similar to Coston et al. [16], with formulations given in Appendix A.2. **(2)** We assume the treatment is monotonic, so $Y(D = 1) \geq Y(D = 0)$ and use the average treatment effect, $\mathbb{E}(Y(D = 1) - Y(D = 0))$, as a sensitivity parameter. For ECPs, there are additional constraints that could be added, for example, if a policy is observed we can include that data explicitly.

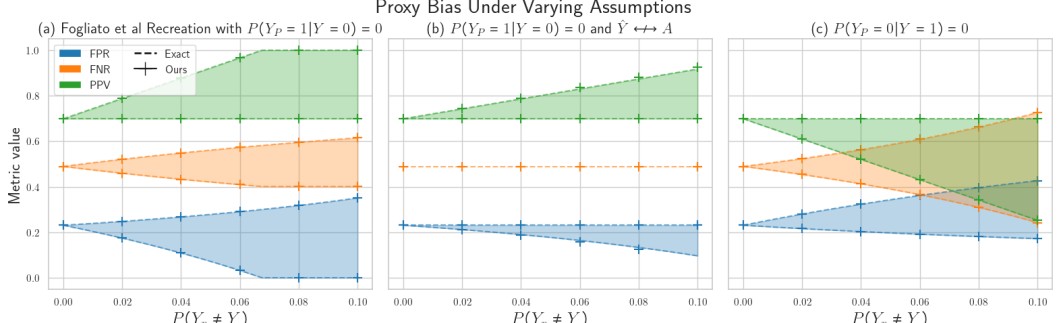

Figure 3: In this we directly recreate the plots from Fogliato et al. [27] for a predictor trained on the COMPAS dataset, allowing for some probabilistic and causal assumptions to vary. The dashed lines represent exact bounds on each statistic for increasing $P(Y_P \neq Y)$, which follow from Fogliato et al. [27] or our derivations in Appendix C.1.2. (a) represents the original setting, where we have $P(Y_P = 1 \mid Y = 0) = 0$, in (b) we drop the dashed edge between $X$ and $Y_P$ in the causal graph in Fig. 2a, and finally for (c) we instead take $P(Y_P = 0 \mid Y = 1) = 0$. As we can see, at all points we query, the automatically derived bounds recover the algebraically derived bounds.

## 6 Experiments

In this section, we showcase the use of causal sensitivity analysis for fairness applications by performing sensitivity analyses for each of the biases introduced above on a set of benchmark datasets. In doing so, we aim to reveal some of the nuances of performing sensitivity analyses for various types of bias. These experiments further lend themselves to some general conclusions about the complexity of real-world fairness evaluations. We present additional results in Appendix C, including tests of causal fairness metrics that reveal the effects of measurement bias in a causal setting.

### 6.1 Recreating Fogliato et al. [27] under varying assumptions

Fogliato et al. [27] aims to assess how sensitive the false positive rate (FPR), false negative rate (FNR), and positive predictive value (PPV) are to proxy bias for a predictor trained on the COMPAS dataset [3], under the assumption that $P(Y_P = 1|Y = 0) = 0$. The outcome in the COMPAS dataset is reported re-offense. This outcome a proxy, because we cannot actually observe whether someone has re-offended, we only know if they were convicted of a new offense. The assumption that $P(Y_P = 1|Y = 0) = 0$ implies that whenever someone is reported to have re-offended they actually re-offend; all the measurement error hence comes from people who did re-offend who did not get caught. We start by recreating the results of the original study from just the DAG in Fig. 8(a). This confirms the correctness of the computational bounds: they always match the true algebraically derived bounds in Fogliato et al. [27].

Next we demonstrate the flexibility of our framework by switching from the assumption that $P(Y_P = 1|Y = 0) = 0$ to $P(Y_P = 0|Y = 1) = 0$. This is probably a less realistic assumption in the COMPAS dataset, implying all measurement error comes from false convictions rather than under-reporting, but in a different context, this might be a more reasonable assumption. We derive the algebraic bounds to confirm the computational bounds match. The point here is that practitioners can easily encode whatever assumptions make sense in their particular context and quickly get results without needing to algebraically derive bounds, and that these results really do vary under different assumptions.

Finally, in Fig. 8(c), we drop the dashed edge between $X$ and $Y_P$ in Fig. 2a. This might seem like an odd experiment: we do not actually observe $X$, and $X$ already causes $Y$, so how could dropping the edge from $X$ to $Y_P$ really matter? Without the $X \to Y_P$ edge $Y_P$ is independent of $U$ conditional on $Y$, which significantly tightens the bounds for FPR and PPV and fully identifies FNR.[2] Once again, we derive the algebraic bounds and find that the computational bounds match. This final experiment simultaneously demonstrates potential drawbacks of causal sensitivity analysis and the enormous benefit of being able to easily run analyses under varying assumptions. Sensitivity analysis can be dangerously misleading if we include unrealistic assumptions, giving us a false sense of security.

---

[2]On a technical level, the identification of the FNR points to the utility of sensitivity analysis for discovering new FairML-specific identification theory, an area which has received little attention to date.

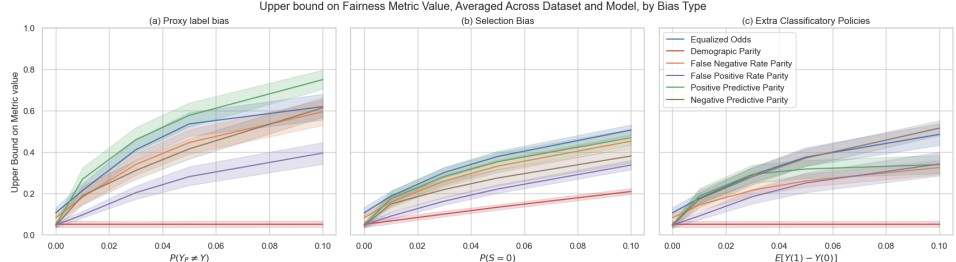

Figure 5: Results of our cross-dataset study, in which we assess the sensitivity of multiple ML predictors trained to satisfy various parity constraints on the fairness benchmarking datasets listed in Appendix D. We can see different metrics are susceptible to bias in different ways, with notably demographic parity being more robust than more complicated, outcome-dependent metrics.

But the remedy is straightforward: analysts should always run an assumption-lite analysis before incorporating more domain-specific information so that they can understand which assumptions are driving their results and make a decision about whether those assumptions are realistic.

## 6.2 Intersection of Biases

Motivated by Section 3.4, we now consider settings exhibiting multiple simultaneous biases. In Fig. 4, we provide a sensitivity plot for proxy and selection bias simultaneously for an equalized odds predictor trained on the *Adult* dataset [7]. We plot the upper bound of the equalized odds value as the sensitivity parameters for both biases vary, with a maximum of 5% proxy labels and 5% of the population being unmeasured. This exercise demonstrates that the presence of multiple biases can quickly render parity evaluation meaningless, with the upper bound reaching the maximum possible value of the statistic. Whilst an upper bound inherently represents the worst-case scenario, this experiment showcases that practitioners must provide additional context-specific assumptions to imbue parity evaluations with meaning in the presence of measurement biases, especially when multiple biases are present simultaneously. We include additional re-

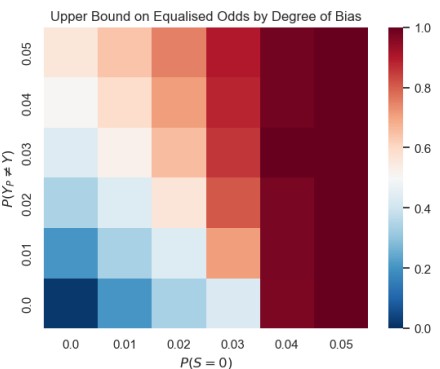

Figure 4: Combination of Proxy and Selection Bias for an equalized odds predictor on the Adult dataset.

sults in Appendix C.2.2, which show that biases compound in unpredictable, nonlinear ways.

## 6.3 Cross-Dataset Experiments

Finally, we leverage our framework to systematically explore the sensitivity of the most ubiquitous parity metrics to the measurement biases here surveyed on real benchmark datasets. We run sensitivity analyses on 14 commonly used fairness datasets from [44] training logistic regression, naïve Bayes, and decision trees to satisfy different parity metrics. In Fig. 5, we provide the results of this experiment, plotting the average upper bound for each metric value at different levels of sensitivity. This further supports our thesis that measurement biases present a severe issue to the informativeness of parity metric evaluation, as we see that even small amounts of bias can render the original parity evaluation meaningless relative to what it seeks to measure. Secondly, we find that this fragility is not uniform across metrics. We can see that demographic parity is, unsurprisingly, more robust than more complicated outcome-dependent metrics. Equalized odds is less robust than FPR/FNR due to the fact that it is measured as the maximum of both. Finally, predictive parity metrics are less robust to biased outcomes than metrics which involve conditioning on the outcome, such as FPR/FNR and equalized odds. We present full experiment details, further analysis, and plots per dataset in Appendix E. We also include a sensitivity for the Folktables dataset [19], which represents one of the most popular datasets for modern FairML.

# 7 Codebase and Web Interface

We have developed both a codebase and a web interface to ensure our framework is as usable as possible. The core of both tools are our bias configs – described in App. F and our codebase documentation—which allow for portable, modular, and reproducible sensitivity analysis. Our codebase is essentially a parser for these configs along with a set of fairness metrics we have implemented. Biases and metrics are designed to allow flexibility to suit particular use cases. The codebase wraps around these biases and fairness metrics and parses them into optimization problems which can be solved to produce bounds, currently using the Autobounds backend. We also provide a website that acts as a user interface for less technical users. The website allows for configs to be loaded or exported, and every element of the config can be edited via the interface. Users can also upload datasets and analyze the sensitivity to their chosen fairness metric/bias combinations.

## Discussion & Limitations

In this work, we have described three prevalent measurement biases, argued that they are almost always present in FairML applications, and put forward a toolkit based on newly available methods in causal inference [21] for understanding how these biases impact parity metric evaluation. To apply this methodology, we have focused on the discrete, oblivious setting, however, causal sensitivity analysis is not limited to this domain [66, 15] and therefore its usefulness to FairML should not be either. We hope that the present work will encourage the causal inference community to look to FairML as a key application area for sensitivity analysis. Finally, additional biases that are likely topical to FairML cannot be readily expressed in this framework. For example, the issue of interference, where individuals within a classified population can affect the outcomes of others, would seem to hold in many FairML applications but can be a challenge to express graphically.

## Acknowledgments

JF gratefully acknowledges funding from the EPSRC. NF thanks the Rhodes Trust for supporting their studies at Oxford, where they conducted a portion of this research. MA is indebted to the Machine Learning Department at Carnegie Mellon University and the ACMI lab for support of this work.

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

# A  Disparity Metric Definitions

## A.1  Observational Metrics

**False Positive Rate Parity**  Definition: $\hat{Y} \perp A \mid Y = 0$

Measured as: $P(\hat{Y} = 1 \mid A = 0, Y = 0) - P(\hat{Y} = 1 \mid A = 1, Y = 0)$

**False Negative Rate Parity**  Definition: $\hat{Y} \perp A \mid Y = 1$

Measured as: $P(\hat{Y} = 1 \mid A = 0, Y = 1) - P(\hat{Y} = 1 \mid A = 1, Y = 1)$

**Positive Predictive Parity**  Definition: $Y \perp A \mid \hat{Y} = 1$

Measured as: $P(Y = 1 \mid A = 0, \hat{Y} = 1) - P(Y = 1 \mid A = 1, \hat{Y} = 1)$

**Negative Predictive Parity**  Definition: $Y \perp A \mid \hat{Y} = 0$

Measured as: $P(Y = 1 \mid A = 0, \hat{Y} = 1) - P(Y = 1 \mid A = 1, \hat{Y} = 0)$

**Equalized Odds**  Definition: $Y \perp A \mid \hat{Y}$

Measured as: $\max\left\{\mathrm{FPR}(Y, A, \hat{Y}), \mathrm{FNR}(Y, A, \hat{Y})\right\}$ for false positive rate (FPR) and false negative rate (FNR) given above.

## A.2  ECP Parity Metric Definitions

**Counterfactual False Positive Rate Parity**  Definition: $\hat{Y} \perp A \mid Y(D = 1) = 0$

Measured as: $P_{\mathcal{C}}(\hat{Y} = 1 \mid A = 0, Y(D = 1) = 0) - P(\hat{Y} = 1 \mid A = 1, Y(D = 1) = 0)$

**Counterfactual False Negative Rate Parity**  Definition: $\hat{Y} \perp A \mid Y = 1$

Measured as: $P(\hat{Y} = 1 \mid A = 0, Y(D = 1) = 1) - P(\hat{Y} = 1 \mid A = 1, Y(D = 1) = 1)$

**Counterfactual Positive Predictive Parity**  Definition: $Y(D = 1) \perp A \mid \hat{Y} = 1$

Measured as: $P(Y(D = 1) = 1 \mid A = 0, \hat{Y} = 1) - P(Y(D = 1) = 1 \mid A = 1, \hat{Y} = 1)$

**Counterfactual Negative Predictive Parity**  Definition: $Y(D = 1) \perp A \mid \hat{Y} = 0$

Measured as: $P(Y(D = 1) = 1 \mid A = 0, \hat{Y} = 1) - P(Y(D = 1) = 1 \mid A = 1, \hat{Y} = 0)$

**Counterfactual Equalised Odds**  Definition: $Y(D = 1) \perp A \mid \hat{Y}$

Measured as: $\max\left\{\mathrm{CF_F PR}(Y, A, \hat{Y}), \mathrm{CF_F NR}(Y, A, \hat{Y})\right\}$ for counterfactual false positive rate ($\mathrm{CF_F PR}$) and counterfactual false negative rate ($\mathrm{CF_F NR}$) given above.

# B  Technical Description

## B.1  Structural Causal Model definition

**Definition 1.** *A **structural causal model (SCM)** over the variables $\mathbf{V} = \{V_1, \cdots, V_d\}$ with latent variables $\mathbf{U} = (U_1, \cdots, U_k)$ consists of a set of structural assignments so that every variable $V_i \in \mathbf{V}$ can be written as:*

$$f_{V_i}(\mathrm{Pa}(V_i)), \quad i = 1, ..., d,$$

*Where $\mathrm{Pa}(V_i) \subset \mathbf{V} \cup \mathbf{U}$ are the **parents** of $V_i$ respectively, $f_{V_i}$ is the **structural equation** for $V_i$, and we have a distribution $P(\mathbf{U})$ which we assume factorises as $P(\mathbf{U}) = \prod_{i=1}^{k} P(U_i)$. The **causal graph**, $\mathcal{G}$, arising from the structural causal model consists of a vertex for each variable in $\mathbf{V} \cup \mathbf{U}$, and an edge $Z \to V$ if $Z \in \mathrm{Pa}(V)$ for $Z \in \mathbf{V} \cup \mathbf{U}$ and $V \in \mathbf{V}$; we assume throughout that $\mathcal{G}$ is acyclic. Finally, letting $\mathcal{F}$ be the set of structural equations, we can denote a causal model by $\mathcal{C} = (\mathcal{F}, P(\mathbf{U}))$ and the set of all causal models over $\mathbf{V}$ as $\mathbb{M}_{\mathbf{V}}$.*

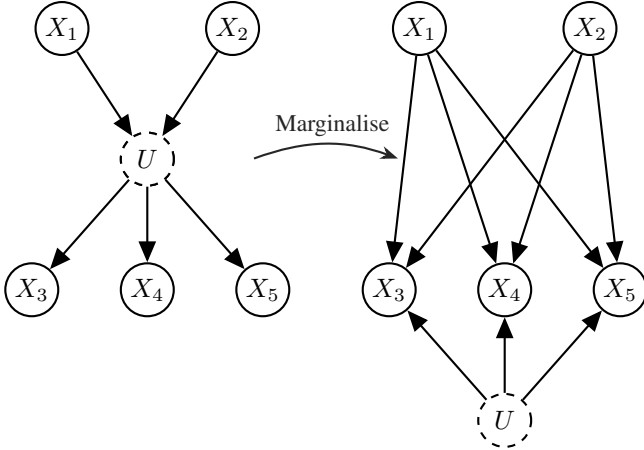

Figure 6: Example of step one in the marginalization, taken from Evans [23].

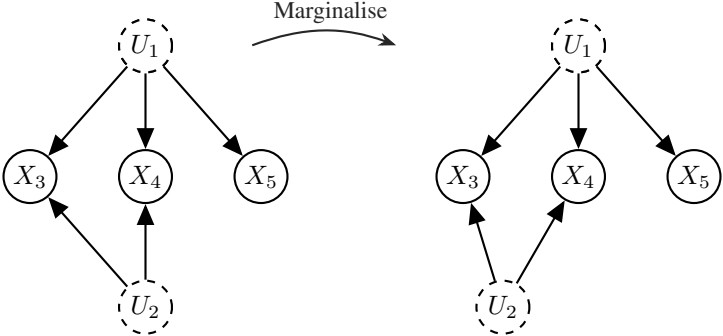

Figure 7: Example of step two in the marginalization.

From the structural causal model we get the *observational distribution*, $P(\mathbf{V})$, by propagating the noise through the structural equations. The *potential outcomes* (counterfactuals) when intervening on a set $\mathbf{A} \subset \mathbf{V}$ are defined via recursive substitution [56], so that when intervening to set $\mathbf{A} = \mathbf{a}$ we take $U_j(\mathbf{a}) = U_j$ and $\mathbf{A}(\mathbf{a}) = \mathbf{a}$ and then defining the general potential outcome as $V_i(\mathbf{a}) := f_{V_i}(\{Z(\mathbf{a})|Z \in \mathrm{Pa}(V_i)\})$. For any event $\star$ involving factual or counterfactual versions of variables in $\mathbf{V}$, we use $P_{\mathcal{C}}(\star)$ to denote the probability of this event under the structural causal model $\mathcal{C}$.

### B.2 Margnalisation in DAGs

**Marginalisation Operation** Suppose $\mathbf{V}$ can be split as $\mathbf{V} = \tilde{\mathbf{V}} \cup \tilde{\mathbf{U}}$ where we are interested in the causal structure over $\tilde{\mathbf{V}}$ and do not observe the variables $\tilde{\mathbf{U}}$. We start from a causal graph $\mathcal{G}$, with unobserved $\tilde{\mathbf{U}}$ we marginalise to get to a graph $\mathcal{G}'$ which is of the form of Definition 1 by doing the following:

1. For all $U \in \tilde{\mathbf{U}}$, add an edge $Z \to \tilde{Z}$ if the current graph contains $Z \to U \to \tilde{Z}$ and then delete any edges $Z \to U$,

2. After completing the first step for all variables in $\tilde{\mathbf{U}}$, delete any $U$ if there exists another $\tilde{U} \in \tilde{\mathbf{U}}$ that influences all of the variables $U$ influences.

Evans [23] showed that there is a structural causal model over the resulting graph which preserves the causal structure over the variables $\tilde{\mathbf{V}}$. Importantly, due to the deletion step, this model has a bounded number of unobserved variables, regardless of how large the set $\tilde{U}$ is.

**Graphical examples**

## B.3 Alternative Causal Graphs for Proxy Bias

Here we provide the following result demonstrating that a wide variety of graphs can give the same outcome under proxy bias:

**Proposition 1.** *So long as any additional unobserved variables $U'$ satisfy the following:*

1. *$U'$ does not cause $A$.*

2. *There is no direct arrow from $U'$ to $\hat{Y}$.*

*Then marginalizing over $U'$ will lead to the same graph as Fig. 2a.*

*Proof.* To show this we need to demonstrate that once we have performed the marginalization operations, no additional edges or nodes will be added to the graph. We do this step by step:

1. This step will add edges if we have two vertices $V, V'$ such that $V \rightarrow U' \rightarrow V'$. However, if neither of $V, V'$ are $\hat{Y}$ then these vertices will already be adjacent in the graph. As the graph is acyclic that means we cannot add any edges.

2. After removing all edges in step one, we will be left so that $U'$ has no parents and affects a subset of vertices in the graph. However, as $U'$ does not cause $A$, this must be a subset of $\{\hat{Y}, Y_P, Y\}$. As these are the vertices caused by $U$ this will lead to the deletion of $U'$.

$\square$

# C   Additional Results

## C.1   Proxy Label Results

### C.1.1   Plots from Fogliato et al. [27] under varying assumptions

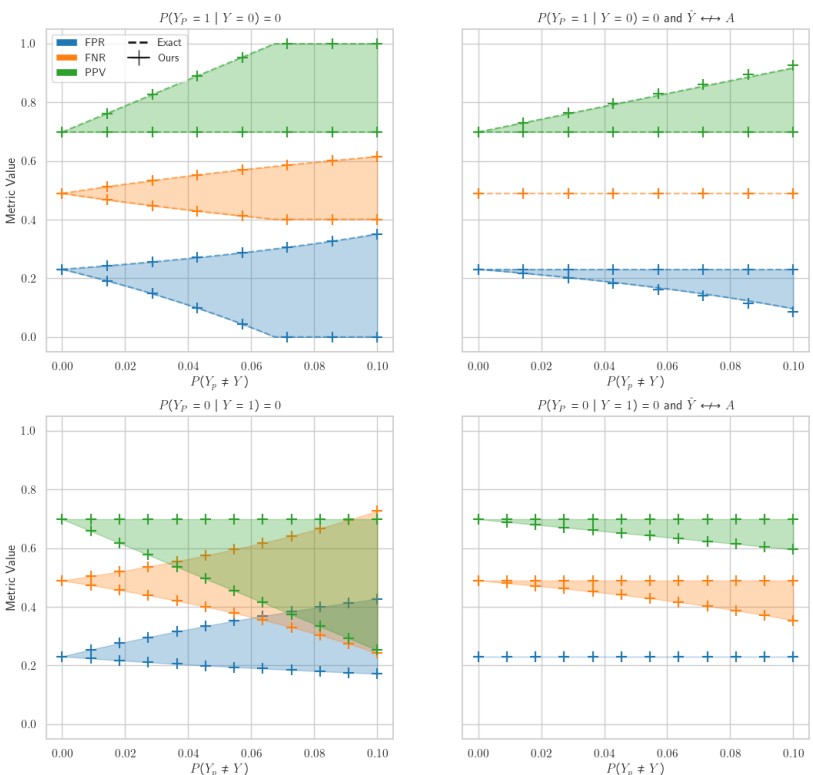

Figure 8: In this plot, we recreate the results from Fogliato et al. [27], where we are interested in the false positive rate (FPR), false negative rate (FNR), and positive predictive value (PPV) for a classifier trained on the COMPAS dataset. In this plot, we consider varying for which $j$ we have $P(Y_P = 1 - j \mid Y = j)$, and we can see that doing so greatly changes the shape of the sensitivity set. Moreover, when we pair these assumptions by dropping of the red dashed edge in Fig. 2a we see we can identify some of the metrics of interest under any degree of bias. For $j = 1$ we identify the FNR and for $j = 0$ we identify the FPR. We prove these identification results in Appendix C.1.2 .

### C.1.2   Proxy Identification Results

In the setup of Fogliato et al. [27], the objective is to obtain the false positive/negative rate in a group $A = a$, where it is assumed that $P(Y = 1, Y_P = 0) = 0$. Now declaring the following parameters:

$$p_{ij} = P(Y_P = i, \hat{Y} = j \mid A = a)$$
$$\alpha_j = P(Y = 1, Y_P = 0, \hat{Y} = j \mid A = a)$$
$$\alpha = \alpha_0 + \alpha_1$$

Under these assumptions $\alpha_0, \alpha_1$ are sufficient to parameterise the distribution, $P(Y, Y_P, \hat{Y} \mid A = a)$. Now, following [27] we have that:

$$\text{FPR}_Y = \frac{p_{01} - \alpha_1}{p_{00} + p_{01} - \alpha}$$
$$\text{FNR}_Y = \frac{p_{10} + \alpha_0}{p_{10} + p_{11} + \alpha}$$
$$\text{PPV}_Y = \frac{p_{11} + \alpha_1}{p_{01} + p_{11}}$$

Now, with the absence of the dashed edge, the DAG in Fig. 2a implies the independence $\hat{Y} \perp Y_P \mid Y, A$. Therefore we get the following:

$$\alpha_j = P(Y = 1, Y_P = 0, \hat{Y} = j \mid A = a)$$

$$= \frac{P(Y = 1, Y_P = 0 \mid A = a)P(Y = 1, \hat{Y} = j \mid A = a)}{P(Y = 1 \mid A = a)}$$

$$= \frac{\alpha(p_{1j} + \alpha_j)}{p_{10} + p_{11} + \alpha}$$

Solving for $\alpha_j$, we get $\alpha_j = \alpha\left(\frac{p_{1j}}{p_{10}+p_{11}}\right)$. Now, inputting this for $\alpha_0$ in the expression for $\mathrm{FNR}_Y$ we get:

$$\mathrm{FNR}_Y = p_{10}\left(\frac{1 + \frac{\alpha}{p_{10}+p_{11}}}{p_{10} + p_{11} + \alpha}\right)$$

$$= \frac{p_{10}}{p_{10} + p_{11}}$$

$$= \mathrm{FNR}_{Y_P}$$

Therefore, under the assumptions given, the true false negative rate is identified and equal to the observed false negative rate on the proxy labels. Inputting the value for $\alpha_1$ into $\mathrm{FPR}_Y$ we instead get:

$$\mathrm{FPR}_Y = \frac{p_{01} - \alpha\left(\frac{p_{10}}{p_{10}+p_{11}}\right)}{(p_{00} + p_{01} - \alpha)}$$

As this is a decreasing function of $\alpha$ we can see that for $\alpha \leq \alpha_0$, $\mathrm{FPR}_Y$ is bounded as:

$$\frac{p_{01}}{(p_{00} + p_{01})} \leq \mathrm{FPR}_Y \leq \frac{p_{01} - \alpha\left(\frac{p_{10}}{p_{10}+p_{11}}\right)}{(p_{00} + p_{01} - \alpha)}$$

For PPV, we again input $\alpha_1$ to give:

$$\mathrm{PPV}_Y = \frac{p_{11} + \alpha\left(\frac{p_{11}}{p_{10}+p_{11}}\right)}{p_{01} + p_{11}}$$

Leading to the bounds:

$$\mathrm{PPV}_{Y_P} \leq \mathrm{PPV}_Y \leq \frac{p_{11} + \alpha\left(\frac{p_{11}}{p_{10}+p_{11}}\right)}{p_{01} + p_{11}}$$

The statements for the identification of the false positive rate and false negative rate are as follows:

**Proposition 2.** *Suppose we have $P(Y_P = 1 \mid Y = 0) = 0$. Then under the conditional independence statement $\hat{Y} \perp Y_P \mid Y, A$, for all level of proxy bias $P(Y_P \neq Y)$:*

$$\mathrm{FNR}_{Y|A=a} = \mathrm{FNR}_{Y_P|A=a}$$

*Where $\mathrm{FNR}_{Y|A=a}$ is the true false negative rate for the group $A = a$ and $\mathrm{FNR}_{Y_P|A=a}$ is the proxied false negative rate.*

*Proof.* Follows from the above derivations. □

Now the equivalent statement for the false positive ratio:

**Proposition 3.** *Suppose we have $P(Y_P = 0 \mid Y = 1) = 0$. Then under the conditional independence statement $\hat{Y} \perp Y_P \mid Y, A$, for all level of proxy bias $P(Y_P \neq Y)$:*

$$\mathrm{FPR}_{Y|A=a} = \mathrm{FPR}_{Y_P|A=a}$$

*Where $\mathrm{FPR}_{Y|A=a}$ is the true false negative rate for the group $A = a$ and $\mathrm{FPR}_{Y_P|A=a}$ is the proxied false negative rate.*

*Proof.* This follows from considering the distribution where $Y, Y_P$ and $\hat{Y}$ are all flipped as any statement about the false positive rate in the original distribution translates to a statement about the false negative rate in the flipped distribution. The assumption $P(Y_P = 0 \mid Y = 1)$ in the original distribution translates to $P(Y_P = 1 \mid Y = 0)$ in the flipped distribution, whereas all other assumptions are symmetric to the flipping operation. Therefore we can apply proposition 2 to see that the flipped FNR is constant under any degree of proxy noise. This leads us to conclude that under these assumptions the FPR in the original distribution must also be constant under any degree of proxy noise. $\qquad\square$

## C.2   Selection Results

### C.2.1   Selective labels under MNAR

Here we include an experiment applying the framework to selective labels under the missing not at random assumption (MNAR)[57]. This supposes that we only see the outcome on a subset of the full dataset, with the outcome on the rest of the dataset free to vary arbitrarily. We work with the Dutch census dataset Van der Laan [61], first fitting an unconstrained logistic regression, then forming the selected population as those who have a predicted probability higher than $0.3$.

Once we have formed the selected subset, we then train four classifiers, each to satisfy a different parity metric. We train to false negative rate parity, false positive rate parity, positive predictive parity, and negative predictive parity. False negative/positive rate parity are trained using the reductions approach [2], whereas for positive predictive parity and negative predictive parity, we train 100 predictors, each weighting different parts of the distribution, taking the one with the lowest parity score above a given accuracy threshold. The plots are shown in Fig. 9.

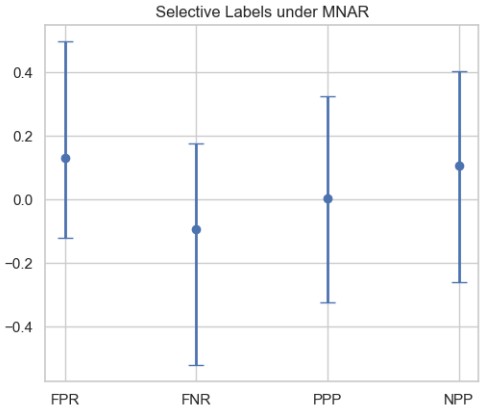

Figure 9: This plot demonstrates a sensitivity analysis for selective labels on the Dutch dataset under the missing, not at random assumption.

### C.2.2   Selection and Proxy Plots

Here we demonstrate the effect of selection and proxy bias jointly on the Adult dataset. We include the results in Fig. 10, which show that the occurrence of multiple biases acts differently for different parity metrics.

## C.3   ECP bias results

### C.3.1   ECP experimental set up

For this experiment, we focus on finding the possible ranges for the counterfactual parity metrics from the given observational statistics. We use the sensitivity parameter of $P(Y(1) \neq Y(0))$, adding additional causal assumptions such as monotonicity ( $Y(1) \geq Y(0)$ ) and if the policy is observed or not. When simulating the policy, we draw $\text{ECP} \sim \text{Ber}(\frac{1}{2} + c * A)$ for $c = 0.2$ to skew the policy in one direction. Results show in Fig. 11

## C.4   Causal Fairness Experiments

In this section, we deliver some results on applying our sensitivity analysis framework to causal fairness metrics of the variety detailed in [50]. Before doing so, we add some technical comments

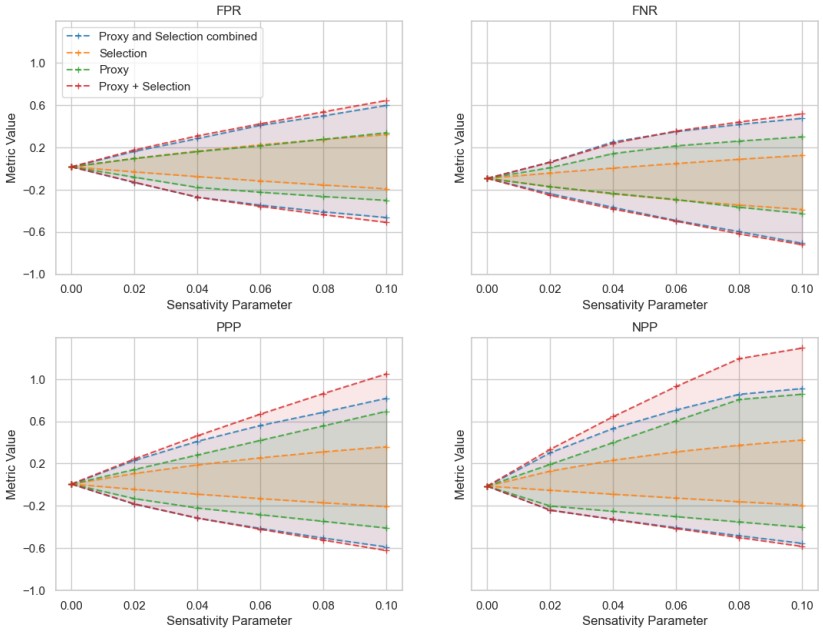

Figure 10: In these plots, we can see the effect of doing a sensitivity analysis jointly for selection and proxy bias. We can see that for the false positive rate parity (FPR) and false negative rate parity (FNR) the combined biases behave roughly as the sum of both biases, however, for positive predictive parity (PPP) and negative predictive parity (NPP) the combination behaves differently with the combined bias amounting to a smaller possible range for the metrics than the sum of the range of both biases individually.

.

on these types of interventions in FairML and some nuances of measurement bias in the context of causal inference.

Firstly, we would like to note that our framework can still be applied to perform sensitivity analysis for measurement bias for other fairness metrics *without* having to consider counterfactuals relative to protected characteristics such as race or gender, thereby avoiding difficulties with intervention on such traits [40, 33, 37]. In this case, $A$ could be seen as denoting membership to a group and indexing different graphs for each group as in Bright et al. [10]. In this case, the arrows leading from $A$ would only express conditional independence relationships as opposed to causal ones. Notably, in the graphs, we suggest they are unconstrained.

Secondly, measurement biases and, specifically, selection bias in causal fairness entail additional complications. This is because almost always, membership in such a dataset is causally downstream of the protected attribute, meaning that when conditioning on individual presence in a dataset, we are introducing selection bias in some form. As Fawkes et al. [26] argue, this means that DAG models will be unable to correctly capture the causal structure in most datasets we come across in FairML. Failing to account for such effects can lead to erroneous causal conclusions.

Having said this, we will proceed with applying the causal graphs in Fig. 2 to do causal fairness analysis for the following metrics:

**Counterfactual Fairness (CF) [42]**  We measure this as $P_{\mathcal{C}}(\hat{Y}(A = 1) \neq \hat{Y}(A = 0))$ which is equal to $0$ exactly when $\hat{Y}$ is counterfactually fair [25].

**Total Effect (TE) [50]**  Measured as $P_{\mathcal{C}}(\hat{Y}(A = 1)) - P_{\mathcal{C}}(\hat{Y}(A = 0))$.

**Spurious Effect (SE) [50]**  Measured as $P_{\mathcal{C}}(\hat{Y}(A = a)) - P_{\mathcal{C}}(\hat{Y} \mid A = a)$.

Results are shown in Fig. 12, where we have assumed that counterfactual fairness is identified at a particular value. We can see that all causal fairness metrics recover a linear relationship under selection in this context.

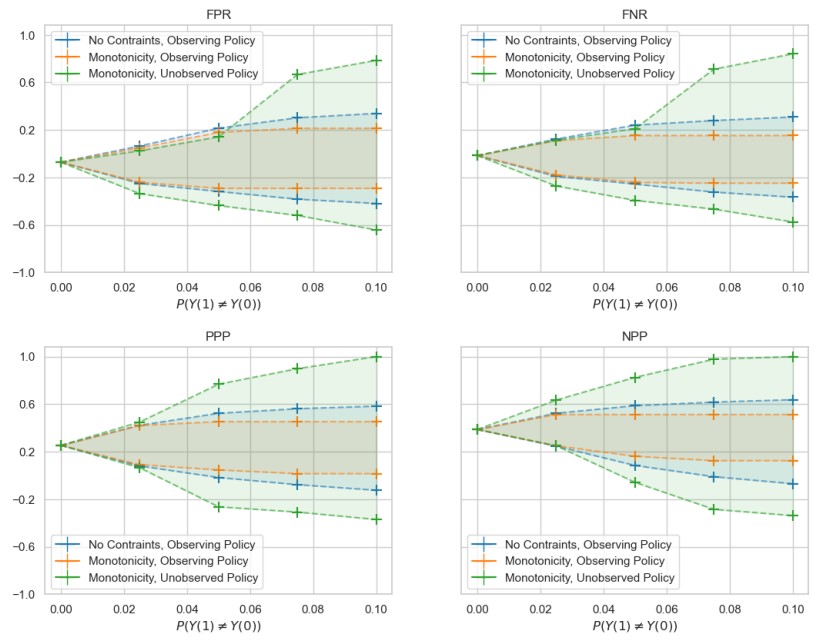

Figure 11: In these plots, we perform a sensitivity analysis for the value of the counterfactual parity metrics given in Appendix A.2. In each case, we work under 3 differing levels of assumption and information

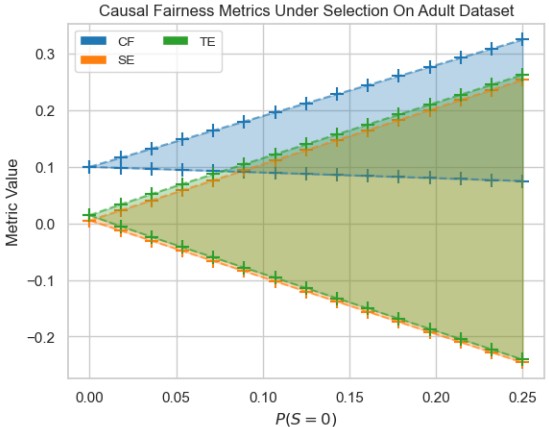

Figure 12: Causal fairness metrics under selection. We show plots for Counterfactual Fairness (CF), Total Effect (TE), and Spurious Effect (SE) using graph 2b where we have additionally assumed that counterfactual fairness is the point identified.

## D   Details of cross dataset bias analysis

In this section, we analyze the datasets presented in Le Quy et al. [44] for the three biases we present in Section 3. We describe each dataset, describe the task that most closely relates to the use of this dataset, and, relative to this task, we describe the measurement biases present. For each bias, we justify our decision.

**Synthetic tasks**   Synthetic tasks are difficult to discuss, since biases are contextual and these tasks are purely theoretical. Given a downstream task, they may or might not exhibit the biases we discuss. We therefore drop them from the analysis.

**Bank Marketing Dataset**   The goal here is to target current clients for the bank to open more accounts. Since the outcome, in this case, is exactly what the bank seeks to maximize, this dataset

| Dataset | Task | Proxy Bias | Selection Bias | ECP Bias |
|---|---|:---:|:---:|:---:|
| Adult | Synthetic | | | |
| KDD Census-Income | Synthetic | | | |
| German credit | Credit risk | | ✓ | ✓ |
| Dutch census | Synthetic | | | |
| Bank marketing | Client | | ✓ | |
| Credit card clients | Default Risk | | ✓ | ✓ |
| COMPAS recid. | Risk prediction | ✓ | ✓ | ✓ |
| COMPAS viol. recid. | Risk prediction | ✓ | ✓ | ✓ |
| Communities&Crime | Neighborhood risk | ✓ | ✓ | ✓ |
| Diabetes | Re-admission risk | ✓ | | ✓ |
| Ricci | Promotion Prediction | ✓ | ✓ | ✓ |
| Student-Mathematics | Admissions | ✓ | ✓ | ✓ |
| Student-Portuguese | Admissions | ✓ | ✓ | ✓ |
| OULAD | Admissions | ✓ | ✓ | ✓ |
| Law School | Admissions | ✓ | ✓ | ✓ |

Table 2: Analysis of the datasets from Le Quy et al. [44], split by task. The explanation for the biases are given in Appendix D.

does not exhibit proxy or ECP bias. However, contacts were made via phone, so there is selection bias in whether customers answered the phone.

**German credit and Credit card clients**    For both of these datasets, the goal is to predict whether customers face default risk. The aim is to use this to decide if applying customers presents a risk to the bank or not. As a result, there will be selection bias, since defaults are only observed for a firms' prior customers. Finally, as with the example in the main text, this exhibits ECP bias, since the firm sets the credit limit, which impacts likelihood of default.

**COMPAS recid. and COMPAS viol. recid. and Communities and Crime**    Datasets built off COMPAS have been well documented to exhibit all these biases and more [5]. Such issues are not unique to COMPAS, and are exhibited in all recidivism and crime prediction datasets. We see similar issues in the Communities and Crime dataset, where the aim is to predict the number of historical crimes per hundred thousand population for a number of cities. Both under-reporting of crimes and over-criminalization render the per-capita crime estimates proxy variables. Due to controversy over the reporting of rape statistics many midwestern communities were excluded, leading to sampling bias. Finally, police practices affect the likelihood of a crime being reported, and police often act in discriminatory ways, so ECP bias seems likely to be a problem as well.

**Diabetes**    For this dataset, the goal is to predict if a patient will be readmitted in the next 30 days. The aim is to use this to assess patient health risk upon leaving the hospital, to determine if they should be discharged. The population is a sample of the patient pool, and so there should not be selection bias. Readmissions differ from the underlying recurring illness, so this does represent a proxy, albeit a fairly reasonable one. In this case, ECP bias is a cause for concern due to the differences in quality of care by demographic group [22].

**Ricci**    The Ricci dataset is an employment dataset, where the goal is to predict the likelihood of a promotion based on a selection of available covariates. A model trained on this data would then be used to predict the potential of applying candidates in order to decide if they are invited to interview or do additional tests. This application would fall prey to all biases we have presented and, as such, strong justification would be required as to the usefulness of the model. Going through biases one by one, proxy bias is exhibited in a similar way to the example presented in the main text, selection bias is present as the model is evaluated on a different population to the one it is trained on and, finally, the firm's policies will have an impact on who succeeds and is promoted at the company.

**Admissions datasets**    The final datasets can all be grouped under admissions to academic institutions. Similarly to the employment example, these are prone to exhibit all the biases we have outlined. This is due to the challenges of having a perfectly objective measure of performance, deployment of models on applying populations but fit on accepted populations, and universities' policies affecting the success of students. Therefore, when using these predictors, arguments should be made as to why using such a measure would not induce demographic skew.

# E  Details of cross-dataset experiment

For this experiment, we train numerous predictors across a variety of common fairness benchmark datasets [44] to satisfy parity constraints. For each dataset we train 18 classifiers total, where the ML model is one of logistic regression, naïve Bayes, and a decision tree and the parity constraint is false negative rate parity, false positive rate parity, positive predictive parity and negative predictive parity, demographic parity, and equalized odds. With the exception of positive/negative predictive parity, we train all classifiers to satisfy these constraints using the reductions approach [2]. For positive/negative predictive parity, we train 100 predictors, each weighting different parts of the distribution, taking the one with the lowest parity score above a given accuracy threshold. We vary the sensitivity parameter over a range of realistic values for many real-world settings, computing the sensitivity bounds for each level of the parameter. We find that, except for demographic parity, all parity measures we evaluate exhibit significant sensitivity over these parameter ranges. This makes it hard to understand what satisfying, e.g., equalised odds, means on a given dataset. The caveat is that equalised odds is only satisfied as long as there are no significant measurement biases in the underlying data, which is almost never the case in FairML audits.

## E.1  Analysis of Results

### E.1.1  Correlational Plots

Here we explore how sensitivity varies according to class imbalance in either $A$ or $Y$. We find that for some metrics (Negative Predictive Parity) class imbalance seems to make little difference to the sensitivity of metrics, with next to no correlation observed between imbalance and sensitivity. This sits in contrast to other metrics (Positive Predictive Parity) where we can see a clear, positive, correlation between class imbalance and sensitivity.

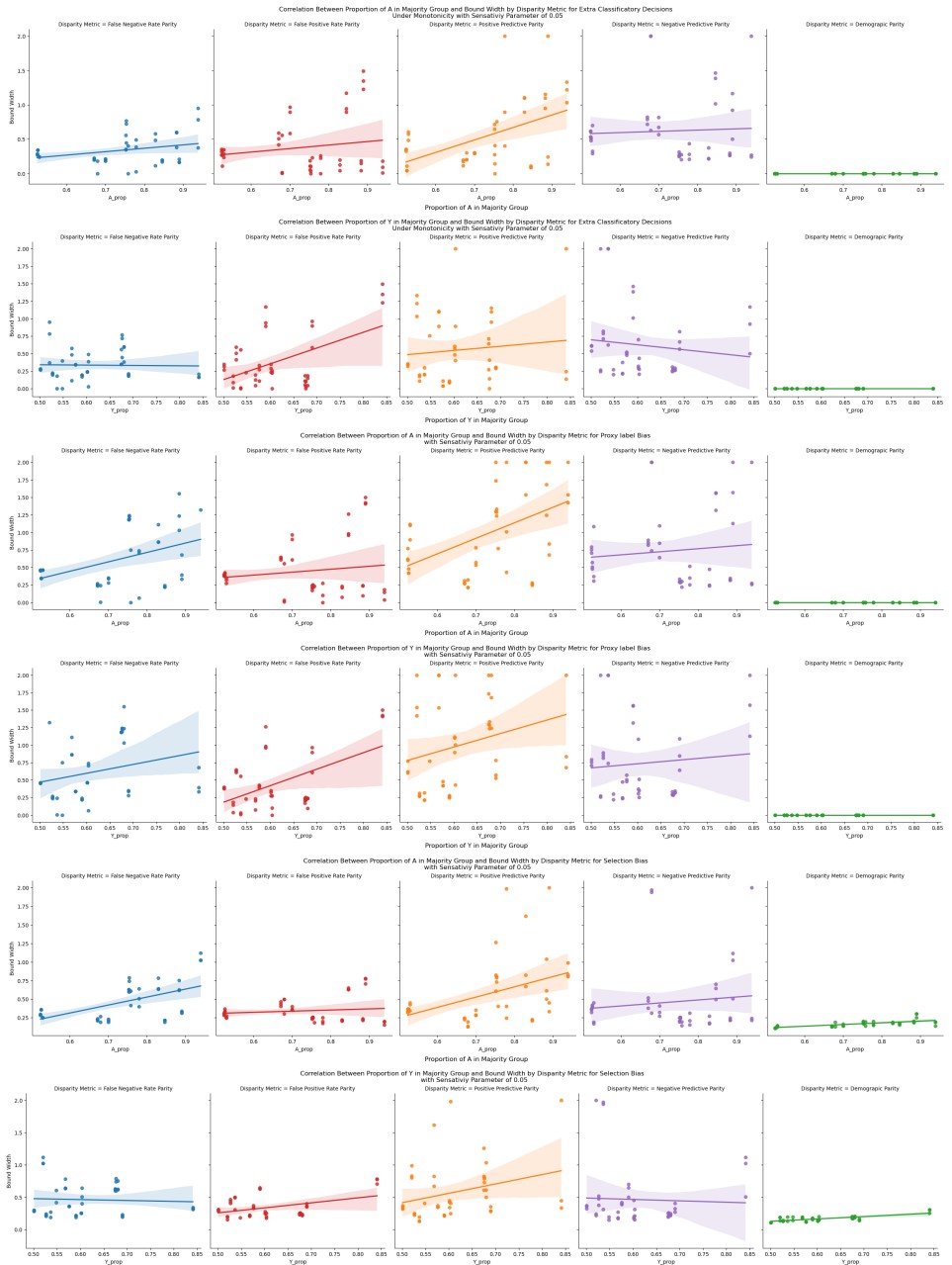

### E.1.2 Cross-Dataset Analysis

These results reveal some heterogeneity across datasets. While we do not find that parity metrics adhere to a universal ordering in terms of robustness against measurement bias, we find that these metrics can be generally grouped together with respect to their order of complexity, and that fragility to bias scales with this complexity. In particular, on the Adult New, Adult, Bank, COMPAS Recid/Viol, Credit, and Dutch datasets, we observe that positive predictive parity is a standout fragile method across biases, followed by false negative rate parity and negative predictive parity. On the German Credit, Law, and Ricci, Student mat/por we see NPP and FNRP tend to be the worst, followed by FPRP and PPP.

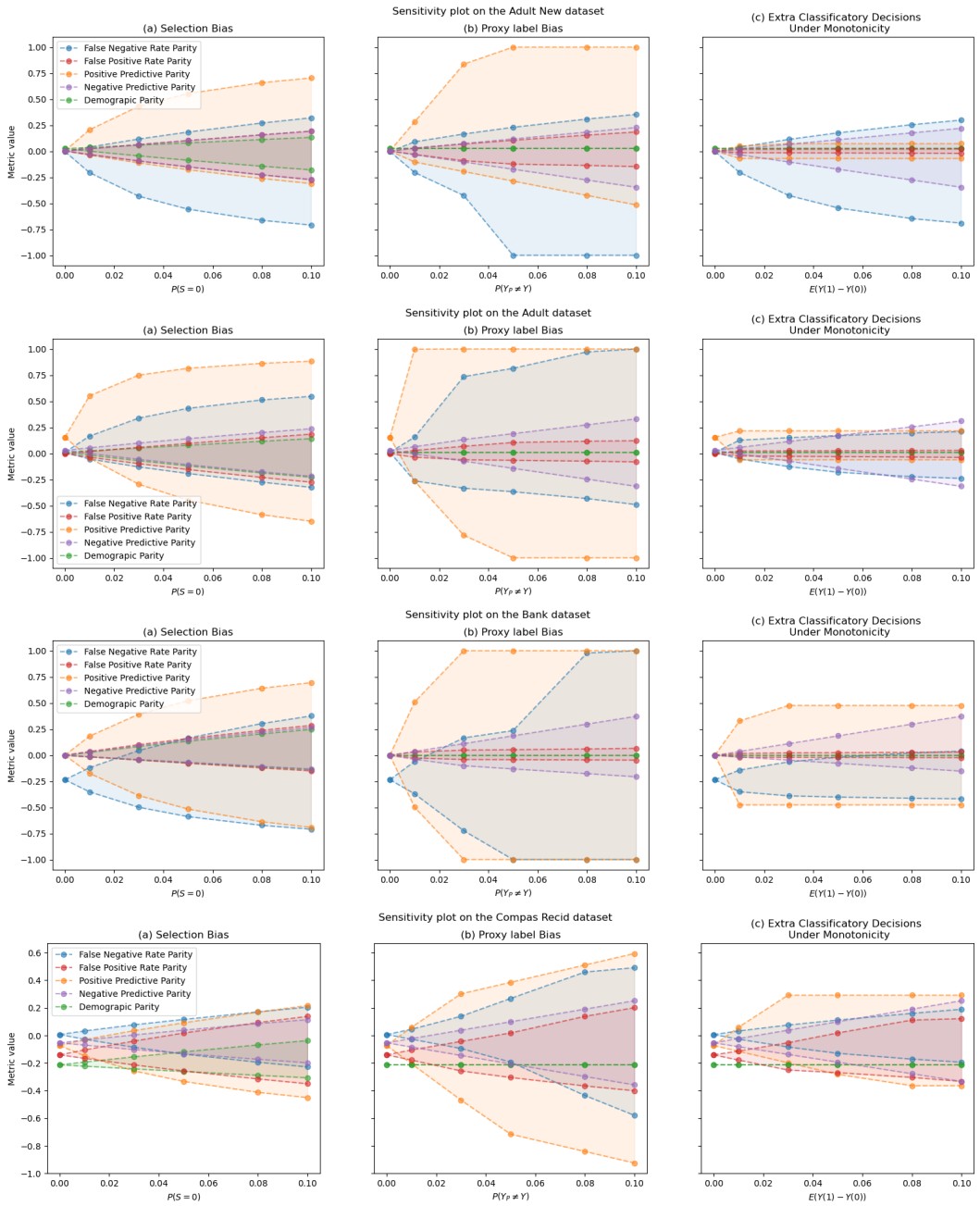

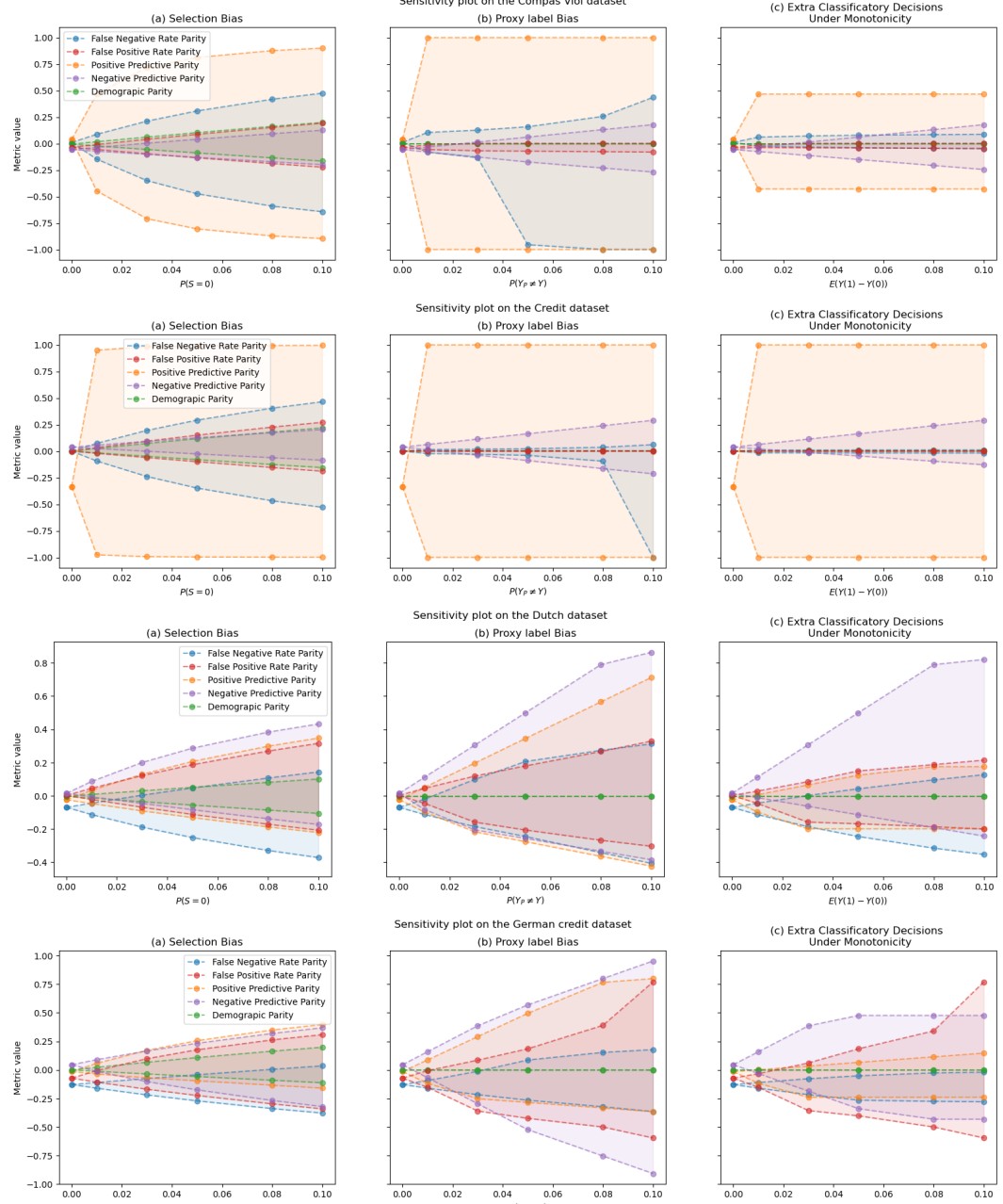

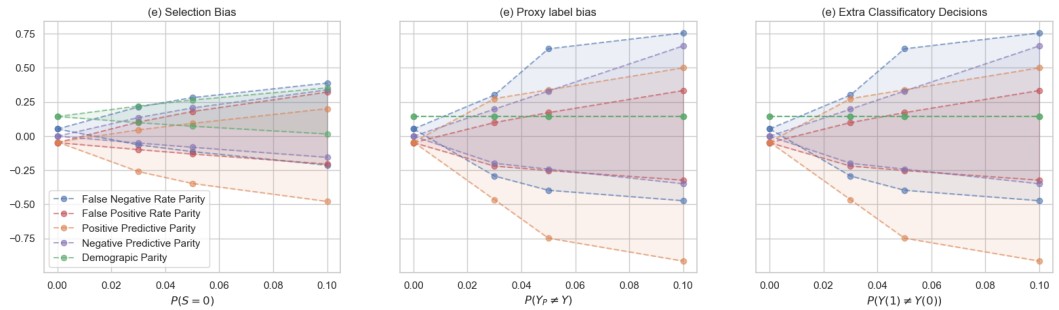

Finally, we also perform a sensitivity analysis for a predictor trained on the folktables [19] dataset:

# F    Codebase and web interface

We have developed both a codebase and a web interface to ensure our framework is as usable as possible. The core of both tools are our bias configs, which allow for portability, modularity, and reproducibility.

The bias config file is a JSON file that specifies the DAG, constraints, and other parameters for the analysis. For example, the selection bias config specifies the following:

```
{
    "dag_str": "A->Y, A->P, A->S, U->P, U->Y, U->S, Y->S",
    "unob": ["U"],
    "cond_nodes": ["S"],
    "attribute_node": "A",
    "outcome_node": "Y",
    "prediction_node": "P",
    "constraints": ["P(S = 1) >= 1 - D"]
}
```

The `dag_str` field is an edgelist for the DAG. The `unob` field specifies the unobserved variables in the DAG, which are used to compute the conditional independencies between the observable variables. The `cond_nodes` field specifies the variables that are observed conditional on some value, e.g., in the selection case we only observe individuals conditional on $S = 1$.

The `attribute_node`, `outcome_node`, and `prediction_node` fields specify the nodes for the observed attribute, outcome, and prediction, which will be used in the parity metric. Usually, these will be set to "A", "Y", and "P" respectively, but some biases may require different nodes. For example, in *Proxy Y Bias*, the observed attribute $Y$ is not the true outcome but a proxy for the true outcome, so the `outcome_node` field would be set to the true outcome node. In the *Proxy Y* config, we call this true outcome $Z$.

The `constraints` field specifies a set of probabilistic constraints on the variables in the DAG. The variable $D$ is a protected variable name that is used to specify the sensitivity parameter, which is a bias-specific level of measurement bias. In the selection bias, we have $P(S = 1) \geq 1 - D$, so $D$ is the probability of observing an individual. In Proxy Y Bias, we have $P(Z = 0 \& Y = 0) + P(Z = 1 \& Y = 1) \geq 1 - D$, so $D$ lower bounds the probability that the observed (proxy) outcome equals the true outcome.

Our codebase is essentially a parser for these configs along with a set of fairness metrics we have implemented. Biases and metrics are designed to allow combinations to suit any particular use case. The codebase wraps around these biases and fairness metrics and parses them into optimization problems which can be solved to produce bounds, currently using the autobounds backend.

The website offers a user interface for constructing and editing bias configs. Configs can be loaded or exported, and every element of the config can be edited via the interface. Users can upload their own datasets and analyze the sensitivity of their dataset to their chosen fairness metric/bias combinations.

# G    Impact Statement

This work aims to broaden the discussion of measurement biases in FairML and provide practical tools for practitioners in the area to use. Our hope is that any potential societal consequences of the work will be positive, corresponding to more equitable algorithmic decision making.

