## A Disparity Metric Definitions

### A.1 Observational Metrics

**False Positive Rate Parity**    Definition: $\hat{Y} \perp A \mid Y = 0$

Measured as: $P(\hat{Y} = 1 \mid A = 0, Y = 0) - P(\hat{Y} = 1 \mid A = 1, Y = 0)$

**False Negative Rate Parity**    Definition: $\hat{Y} \perp A \mid Y = 1$

Measured as: $P(\hat{Y} = 1 \mid A = 0, Y = 1) - P(\hat{Y} = 1 \mid A = 1, Y = 1)$

**Positive Predictive Parity**    Definition: $Y \perp A \mid \hat{Y} = 1$

Measured as: $P(Y = 1 \mid A = 0, \hat{Y} = 1) - P(Y = 1 \mid A = 1, \hat{Y} = 1)$

**Negative Predictive Parity**    Definition: $Y \perp A \mid \hat{Y} = 0$

Measured as: $P(Y = 1 \mid A = 0, \hat{Y} = 1) - P(Y = 1 \mid A = 1, \hat{Y} = 0)$

**Equalized Odds**    Definition: $Y \perp A \mid \hat{Y}$

Measured as: $\max \left\{ \mathrm{FPR}(Y, A, \hat{Y}), \mathrm{FNR}(Y, A, \hat{Y}) \right\}$ for false positive rate (FPR) and false negative rate (FNR) given above.

### A.2 ECP Parity Metric Definitions

**Counterfactual False Positive Rate Parity**    Definition: $\hat{Y} \perp A \mid Y(D = 1) = 0$

Measured as: $P_{\mathcal{C}}(\hat{Y} = 1 \mid A = 0, Y(D = 1) = 0) - P(\hat{Y} = 1 \mid A = 1, Y(D = 1) = 0)$

**Counterfactual False Negative Rate Parity**    Definition: $\hat{Y} \perp A \mid Y = 1$

Measured as: $P(\hat{Y} = 1 \mid A = 0, Y(D = 1) = 1) - P(\hat{Y} = 1 \mid A = 1, Y(D = 1) = 1)$

**Counterfactual Positive Predictive Parity**    Definition: $Y(D = 1) \perp A \mid \hat{Y} = 1$

Measured as: $P(Y(D = 1) = 1 \mid A = 0, \hat{Y} = 1) - P(Y(D = 1) = 1 \mid A = 1, \hat{Y} = 1)$

**Counterfactual Negative Predictive Parity**    Definition: $Y(D = 1) \perp A \mid \hat{Y} = 0$

Measured as: $P(Y(D = 1) = 1 \mid A = 0, \hat{Y} = 1) - P(Y(D = 1) = 1 \mid A = 1, \hat{Y} = 0)$

**Counterfactual Equalised Odds**    Definition: $Y(D = 1) \perp A \mid \hat{Y}$

Measured as: $\max \left\{ \mathrm{CF_F PR}(Y, A, \hat{Y}), \mathrm{CF_F NR}(Y, A, \hat{Y}) \right\}$ for counterfactual false positive rate ($\mathrm{CF_F PR}$) and counterfactual false negative rate ($\mathrm{CF_F NR}$) given above.

## B Technical Description

### B.1 Margnalisation in DAGs

**Marginalisation Operation**    Suppose $\mathbf{V}$ can be split as $\mathbf{V} = \tilde{\mathbf{V}} \cup \tilde{\mathbf{U}}$ where we are interested in the causal structure over $\tilde{\mathbf{V}}$ and do not observe the variables $\tilde{\mathbf{U}}$. We start from a causal graph $\mathcal{G}$, with unobserved $\tilde{\mathbf{U}}$ we marginalise to get to a graph $\mathcal{G}'$ which is of the form of Definition 1 by doing the following:

1. For all $U \in \tilde{\mathbf{U}}$, add an edge $Z \to \tilde{Z}$ if the current graph contains $Z \to U \to \tilde{Z}$ and then delete any edges $Z \to U$,

2. After completing the first step for all variables in $\tilde{\mathbf{U}}$, delete any $U$ if there exists another $\tilde{U} \in \tilde{\mathbf{U}}$ that influences all of the variables $U$ influences.

Evans [22] showed that there is a structural causal model over the resulting graph which preserves the causal structure over the variables $\tilde{\mathbf{V}}$. Importantly, due to the deletion step, this model has bounded number of unobserved variables, regardless of how large the set $\tilde{U}$ is.

**Graphical examples**

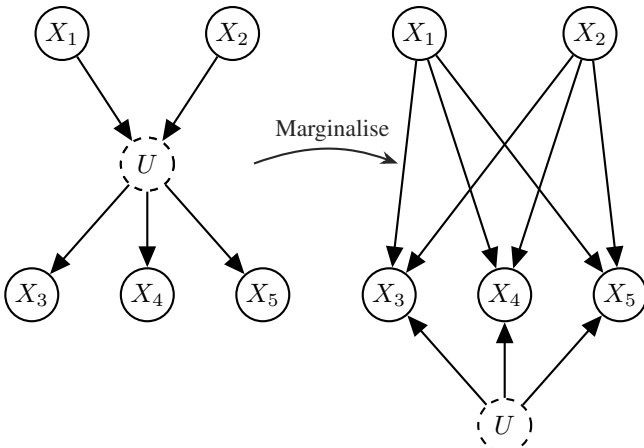

Figure 5: Example of step one in the marginalisation, taken from Evans [22].

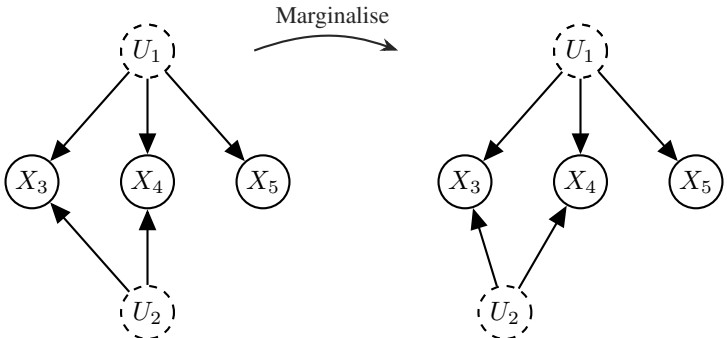

Figure 6: Example of step two in the marginalisation.

### B.2 Alternative Causal Graphs for Proxy Bias

Here we provide the following result demonstrating that a wide variety of Graphs can give the same outcome under proxy bias:

**Proposition 1.** *So long as any additional unobserved variables $U'$ satisfy the following:*

    *1. $U'$ does not cause $A$.*

    *2. There is no direct arrow from $U'$ to $\hat{Y}$.*

*Then marginalising over $U'$ will lead to the same graph as Figure 1a.*

*Proof.* To show this we need to demonstrate that once we have performed the marginalisation operations, no additional edges or nodes will be added to the graph. We do this step by step:

    1. This step will add edges if we have two vertices $V, V'$ such that $V \to U' \to V'$. However, if neither of $V, V'$ are $\hat{Y}$ then these vertices will already be adjacent in the graph. As the graph is acyclic that means we cannot be adding any edges.

    2. After removing all edges in step one, we will be left so that $U'$ has no parents and effects a subset of vertices in the graph. However, as $U'$ does not cause $A$, this must be a subset of $\{\hat{Y}, Y_P, Y\}$. As these are the vertices caused by $U$ this will lead to the deletion of $U'$.

$\qquad\qquad\qquad\qquad\qquad\qquad\qquad\qquad\qquad\qquad\qquad\qquad\qquad\qquad\qquad\qquad\qquad\qquad\qquad\qquad\quad \square$

| Dataset | Task | Proxy Bias | Selection Bias | ECP Bias |
|---|---|:---:|:---:|:---:|
| Adult | Synthetic | | | |
| KDD Census-Income | Synthetic | | | |
| German credit | Credit risk | | ✓ | ✓ |
| Dutch census | Synthetic | | | |
| Bank marketing | Client | | ✓ | |
| Credit card clients | Default Risk | | ✓ | ✓ |
| COMPAS recid. | Risk prediction | ✓ | ✓ | ✓ |
| COMPAS viol. recid. | Risk prediction | ✓ | ✓ | ✓ |
| Communities&Crime | Neighborhood risk | ✓ | ✓ | ✓ |
| Diabetes | Re-admission risk | ✓ | | ✓ |
| Ricci | Promotion Prediction | ✓ | ✓ | ✓ |
| Student-Mathematics | Admissions | ✓ | ✓ | ✓ |
| Student-Portuguese | Admissions | ✓ | ✓ | ✓ |
| OULAD | Admissions | ✓ | ✓ | ✓ |
| Law School | Admissions | ✓ | ✓ | ✓ |

Table 2: Analysis of the datasets from Le Quy et al. [43], split by task. The explanation for the biases are given in Appendix E.

## C Cross Dataset Analysis

In this section we analyse the datasets presented in Le Quy et al. [43] for the three biases we present in Section 3. We describe each dataset, give the task which most closely relates to the use of this dataset, and relative to this task we decide if each of the three measurement biases are present or not. For each bias we provide a justification of our decision.

**Synthetic tasks**   The synthetic tasks are hard to discuss since the biases are contextual and these tasks are purely theoretical. Given a downstream task they might or might not have the biases we discuss. Therefore we drop them from the analysis.

**Bank marketing Dataset**   The goal here is to target current clients for the bank to open more accounts. Since the outcome in this case is exactly what the bank seeks to maximise, this dataset does not exhibit proxy or ECP bias. However, contacts we made via phone, so there is selection bias in whether people answered the phone.

**German credit and Credit card clients**   For both of these datasets goal is to predict whether customers face default risk. The aim is to use this to decide if applying customers present a risk to the bank or not. As a result of this, there will be selection bias due to the fact that since defaults are only observed for the firms' previous customers. Finally as with the example in the main text, this exhibits extra-classificatory policy bias since the firm sets the credit limit which impacts the likelihood of default.

**COMPAS recid. and COMPAS viol. recid. and Communities and Crime**   Datasets build off COMPAS have been well documented to exhibit all these biases and more [5]. These issues are not unique to COMPAS and are exhibited in all other recidivism and crime prediction datasets, as such they will also apply to communities and crime, where the aim is to predict number of historical crimes per hundred thousand population for a number of states. Moreover, a large degree of missing values in this dataset show the issues due to selection bias.

**Diabetes**   For this dataset, the goal is to predict if a patient will be readmitted in the next 30 days. The aim is to use this to decide how much of a health risk a given patient is upon leaving the hospital, to decide if they should be kept there. The population is a sample of the patient pool, and so there should not be selection bias. Readmissions are different from the underling recurring illness, so this does represent a proxy, albeit it a fairly reasonable one. In this case ECP bias is a cause for concern due to the differences in quality of care by demographic group [21].

**Ricci**   The Ricci dataset is an employment dataset, where the goal is to predict the likelihood of a promotion based off of a selection of available covariates. A model trained on this data would then be

used to predict the potential of applying candidates in order to decide if they are invited to interview or do additional tests. This application would fall risk of all the biases we have presented and as such strong justification would be required as to the usefulness of the model. Going through one by one, proxy bias is exhibited in a similar way to the example presented in the main text, selection bias is present as the model is evaluated on a different population to the one it is trained on, and finally the firms policies will have an impact on who succeeds and is promoted at the company.

**Admissions datasets**  The final datasets can all be grouped under admissions to academic institutions. Similarly to the employment example, these will exhibit all the biases we have outlined. This is because of the challenges of having a perfectly objective measure of performance, models being used on applying populations but fit on accepted populations, and the universities policies affecting the success of students. Therefore, when using these predictors arguments should be made about why using such a measure would not induce demographic skew.

# D  Additional Results

## D.1  Proxy Label Results

### D.1.1  Plots from Fogliato et al. [26] under varying assumptions

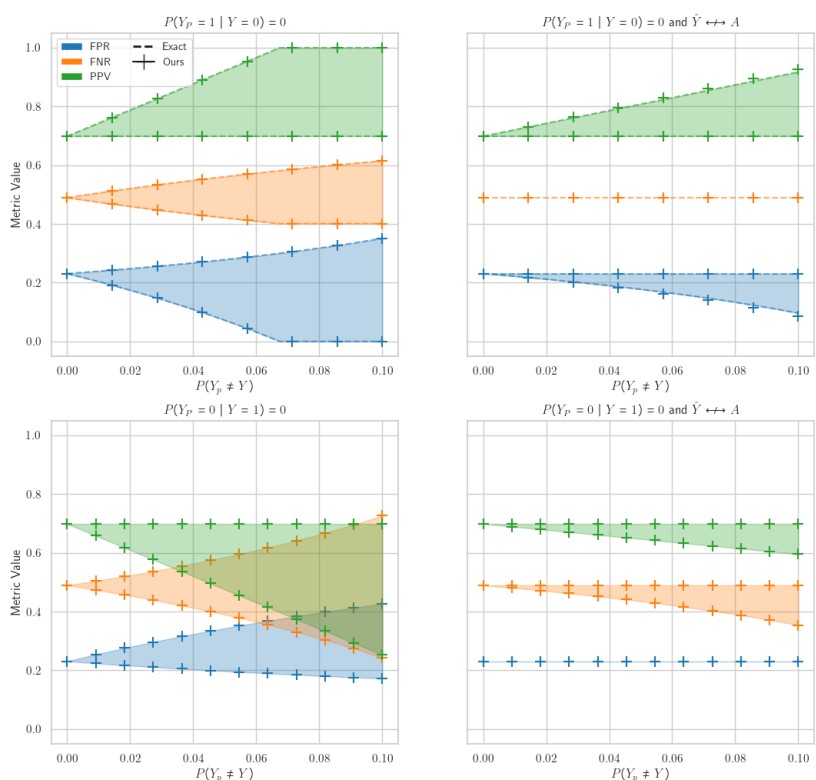

Figure 7: In this plot, we recreate the results from Fogliato et al. [26], where we are interested in the false positive rate (FPR), false negative rate (FNR), and positive predictive value (PPV) for a classifier trained on the COMPAS dataset. In this plot we consider varying for which $j$ we have $P(Y_P = 1 - j \mid Y = j)$, and we can see that doing so greatly changes the shape of the sensitivity set. Moreover, when we pair these assumptions dropping of the red dashed edge in Figure 1a we see we can identify some of the metrics of interest under any degree of bias. For $j = 1$ we identify the FNR and for $j = 0$ we identify the FPR. We prove these identification results in Appendix D.1.2 .

 **D.1.2 Proxy Identification Results**

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

 decide if each of the three measurement biases are present or not. For each bias we provide a justification of our decision.

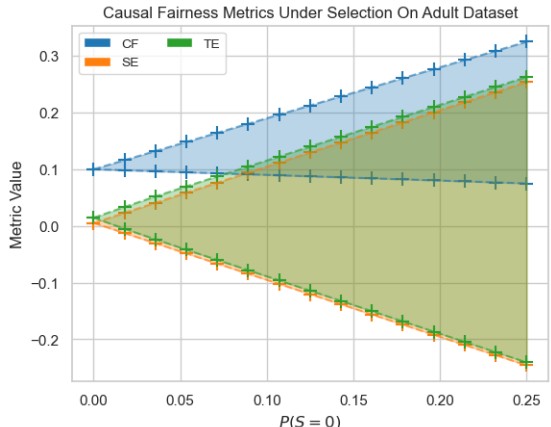

Figure 11: Causal fairness metrics under selection. We show plots for Counterfactual Fairness (CF),Total Effect (TE), and Spurious Effect (SE) using graph 1b where we have additionally assumed that counterfactual fairness is point identified.

| Dataset | Task | Proxy Bias | Selection Bias | ECP Bias |
|---|---|:---:|:---:|:---:|
| Adult | Synthetic | | | |
| KDD Census-Income | Synthetic | | | |
| German credit | Credit risk | | ✓ | ✓ |
| Dutch census | Synthetic | | | |
| Bank marketing | Client | | ✓ | |
| Credit card clients | Default Risk | | ✓ | ✓ |
| COMPAS recid. | Risk prediction | ✓ | ✓ | ✓ |
| COMPAS viol. recid. | Risk prediction | ✓ | ✓ | ✓ |
| Communities&Crime | Neighborhood risk | ✓ | ✓ | ✓ |
| Diabetes | Re-admission risk | ✓ | | ✓ |
| Ricci | Promotion Prediction | ✓ | ✓ | ✓ |
| Student-Mathematics | Admissions | ✓ | ✓ | ✓ |
| Student-Portuguese | Admissions | ✓ | ✓ | ✓ |
| OULAD | Admissions | ✓ | ✓ | ✓ |
| Law School | Admissions | ✓ | ✓ | ✓ |

Table 3: Analysis of the datasets from Le Quy et al. [43], split by task. The explanation for the biases are given in Appendix E.

**Synthetic tasks**  The synthetic tasks are hard to discuss since the biases are contextual and these tasks are purely theoretical. Given a downstream task they might or might not have the biases we discuss. Therefore we drop them from the analysis.

**Bank marketing Dataset**  The goal here is to target current clients for the bank to open more accounts. Since the outcome in this case is exactly what the bank seeks to maximise, this dataset does not exhibit proxy or ECP bias. However, contacts we made via phone, so there is selection bias in whether people answered the phone.

**German credit and Credit card clients**  For both of these datasets goal is to predict whether customers face default risk. The aim is to use this to decide if applying customers present a risk to the bank or not. As a result of this, there will be selection bias due to the fact that since defaults are only observed for the firms' previous customers. Finally as with the example in the main text, this exhibits extra-classificatory policy bias since the firm sets the credit limit which impacts the likelihood of default.

**COMPAS recid. and COMPAS viol. recid. and Communities and Crime**  Datasets build off COMPAS have been well documented to exhibit all these biases and more [5]. These issues are not unique to COMPAS and are exhibited in all other recidivism and crime prediction datasets, as

such they will also apply to communities and crime, where the aim is to predict number of historical crimes per hundred thousand population for a number of states. Moreover, a large degree of missing values in this dataset show the issues due to selection bias.

**Diabetes** For this dataset, the goal is to predict if a patient will be readmitted in the next 30 days. The aim is to use this to decide how much of a health risk a given patient is upon leaving the hospital, to decide if they should be kept there. The population is a sample of the patient pool, and so there should not be selection bias. Readmissions are different from the underling recurring illness, so this does represent a proxy, albeit it a fairly reasonable one. In this case ECP bias is a cause for concern due to the differences in quality of care by demographic group [21].

**Ricci** The Ricci dataset is an employment dataset, where the goal is to predict the likelihood of a promotion based off of a selection of available covariates. A model trained on this data would then be used to predict the potential of applying candidates in order to decide if they are invited to interview or do additional tests. This application would fall risk of all the biases we have presented and as such strong justification would be required as to the usefulness of the model. Going through one by one, proxy bias is exhibited in a similar way to the example presented in the main text, selection bias is present as the model is evaluated on a different population to the one it is trained on, and finally the firms policies will have an impact on who succeeds and is promoted at the company.

**Admissions datasets** The final datasets can all be grouped under admissions to academic institutions. Similarly to the employment example, these will exhibit all the biases we have outlined. This is because of the challenges of having a perfectly objective measure of performance, models being used on applying populations but fit on accepted populations, and the universities policies affecting the success of students. Therefore, when using these predictors arguments should be made about why using such a measure would not induce demographic skew.

# F   Details of cross dataset experiment

For this experiment, we train numerous predictors across a variety of common fairness benchmarking datasets [43] to satisfy parity constraints. For each dataset we train 18 classifiers total, where the model ML is one of logistic regression, naïve Bayes and a decision tree and the parity constraint is false negative rate parity, false positive rate parity, positive predictive parity and negative predictive parity, demographic parity and equalized odds. With the exception of positive/negative predictive parity we train all classifier to satisfy these constraints using the reductions approach [2]. For positive/negative predictive parity, we train 100 predictors, each weighting different parts of the distribution, taking the one with the lowest parity score above a given accuracy threshold. We vary the sensitivity parameter over a range of realistic values for many real-world settings, computing the sensitivity bounds for each level of the parameter. We find that, except for demographic parity, all parity measures we evaluate exhibit significant sensitivity over these parameter ranges. This makes it hard to understand what satisfying, e.x. equalised odds means on a given dataset. The caveat is that equalised odds is only satisfied as long as there are no significant measurement biases in the underlying data, which is almost never the case in FairML audits.

## F.1   Analysis of Results

### F.1.1   Correlational Plots

Here we explore how sensitivity varies according to class imbalance in either $A$ or $Y$. We find that for some metrics (Negative Predictive Parity) class imbalance seems to make little difference to the sensitivity of metrics, with next to no correlation observed between imbalance and sensitivity. This lies in contrast to other metrics (Positive Predictive Parity) where we can see a much more clear, positive, correlation between class imbalance and sensitivity.

### F.1.2   Cross Dataset Analysis

These results unpack some heterogeneity across datasets. We find that there are not hard and fast rules here, each dataset and each bias requires its own analysis. At the same time this is broadly consistent with our central contention that complexity goes hand in hand with fragility. In particular,

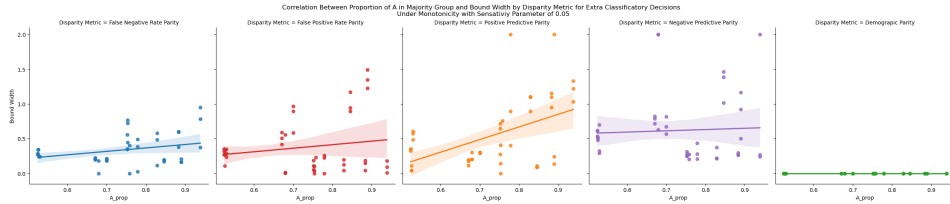

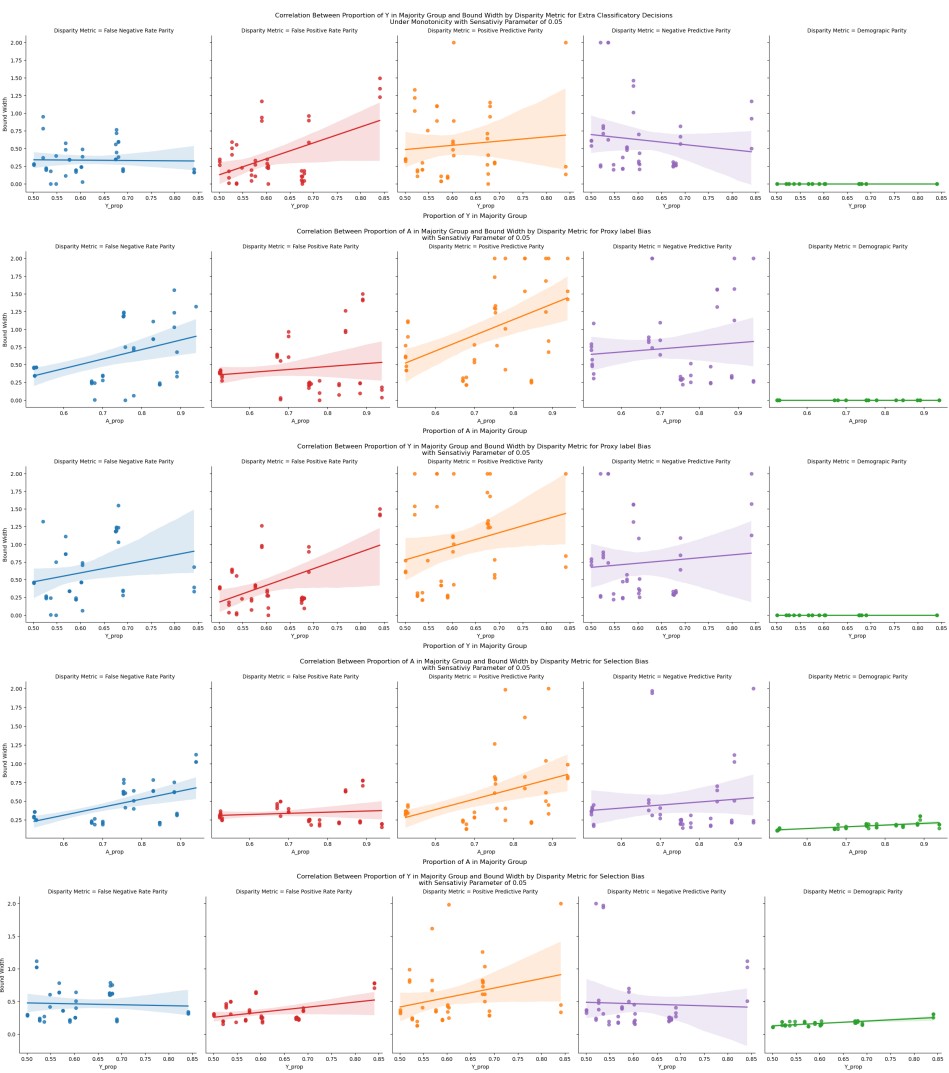

on the Adult New, Adult, Bank, Compas Recid/Viol, Credit, Dutch we see positive predictive parity is
a standout fragile method across biases, followed by false negative rate pairt and negative predictive
parity. On the German Credit, Law, and Ricci, Student mat/por we see NPP and FNRP tend to be the
worst, followed by FPRP and PPP.

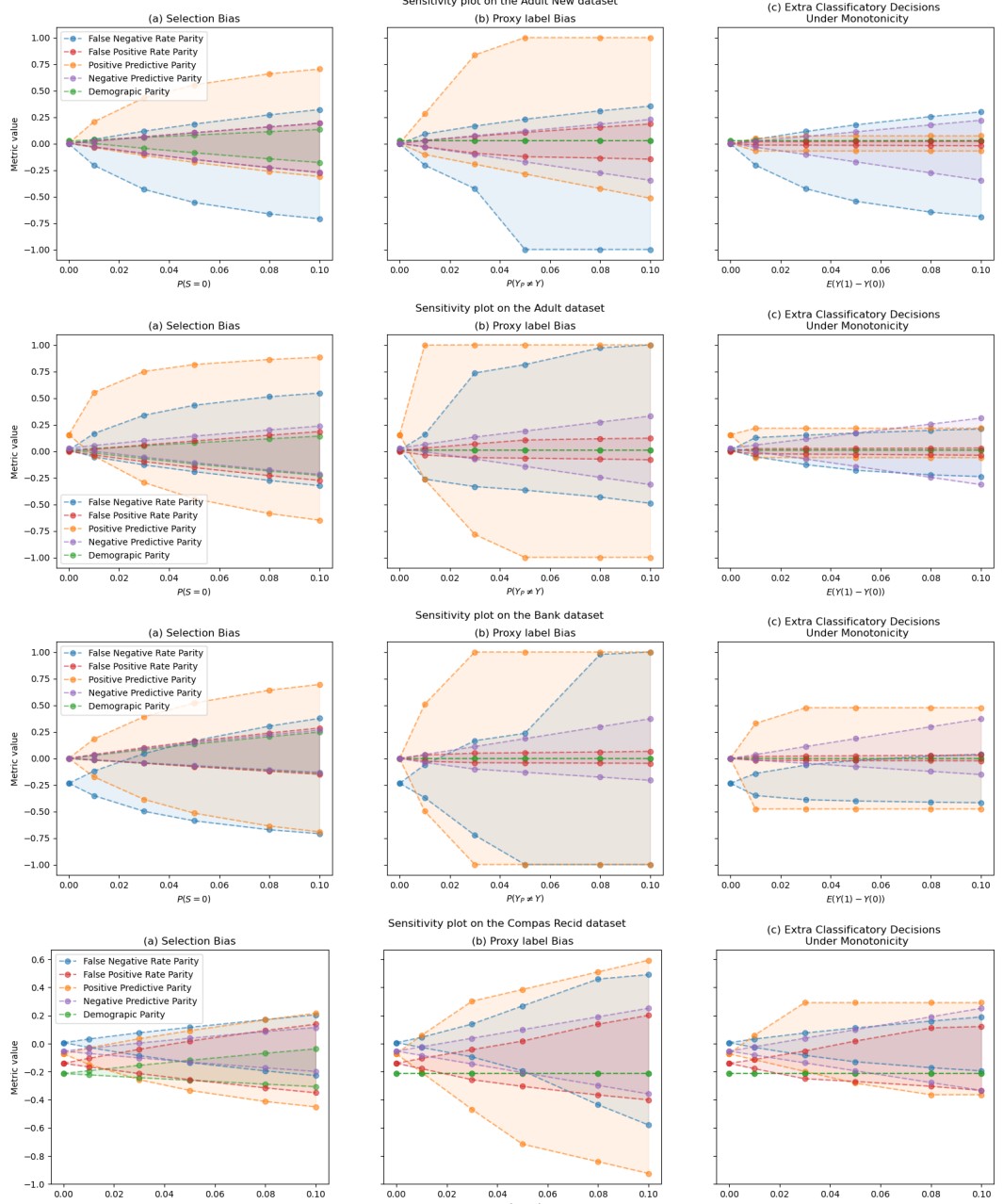

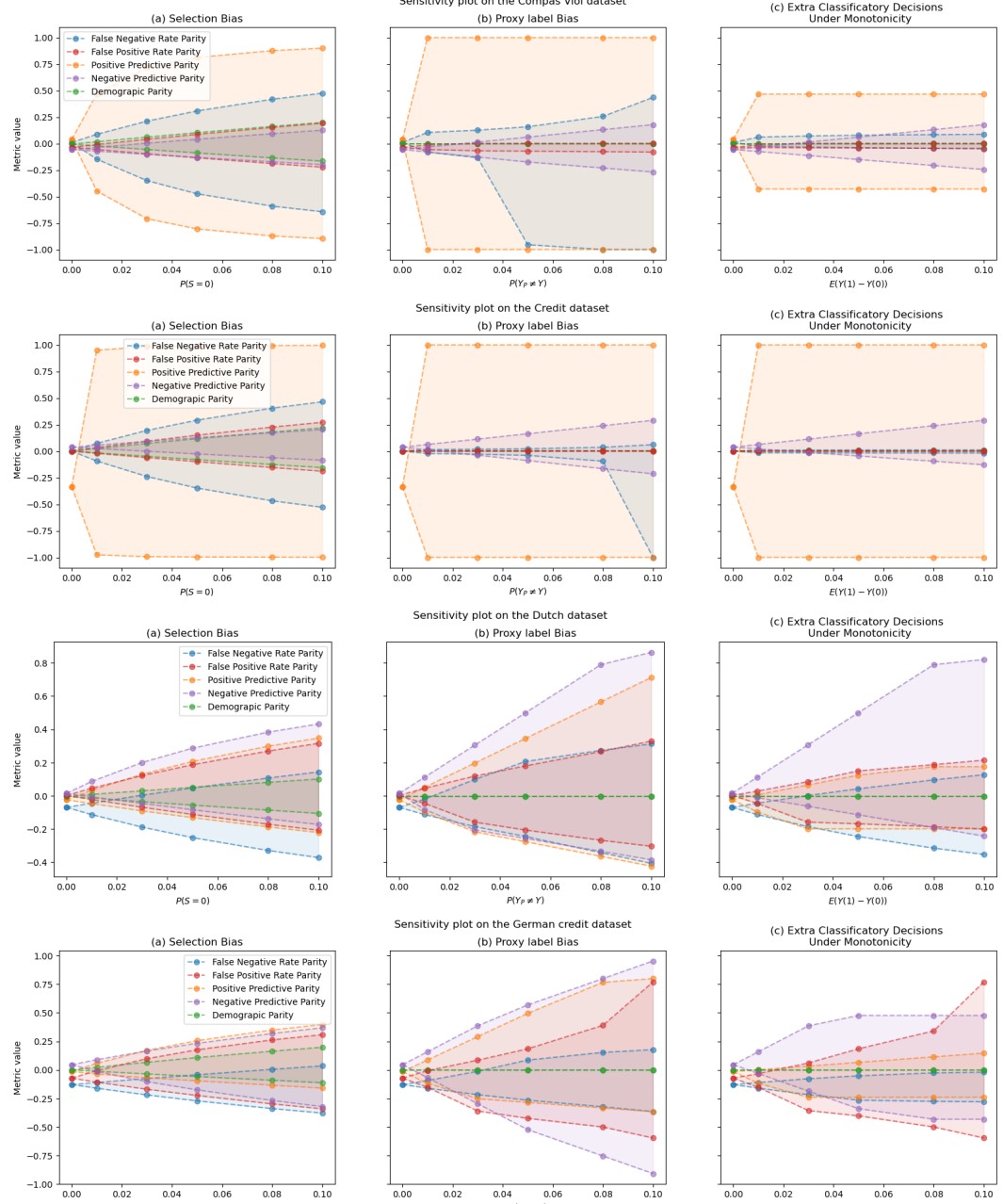

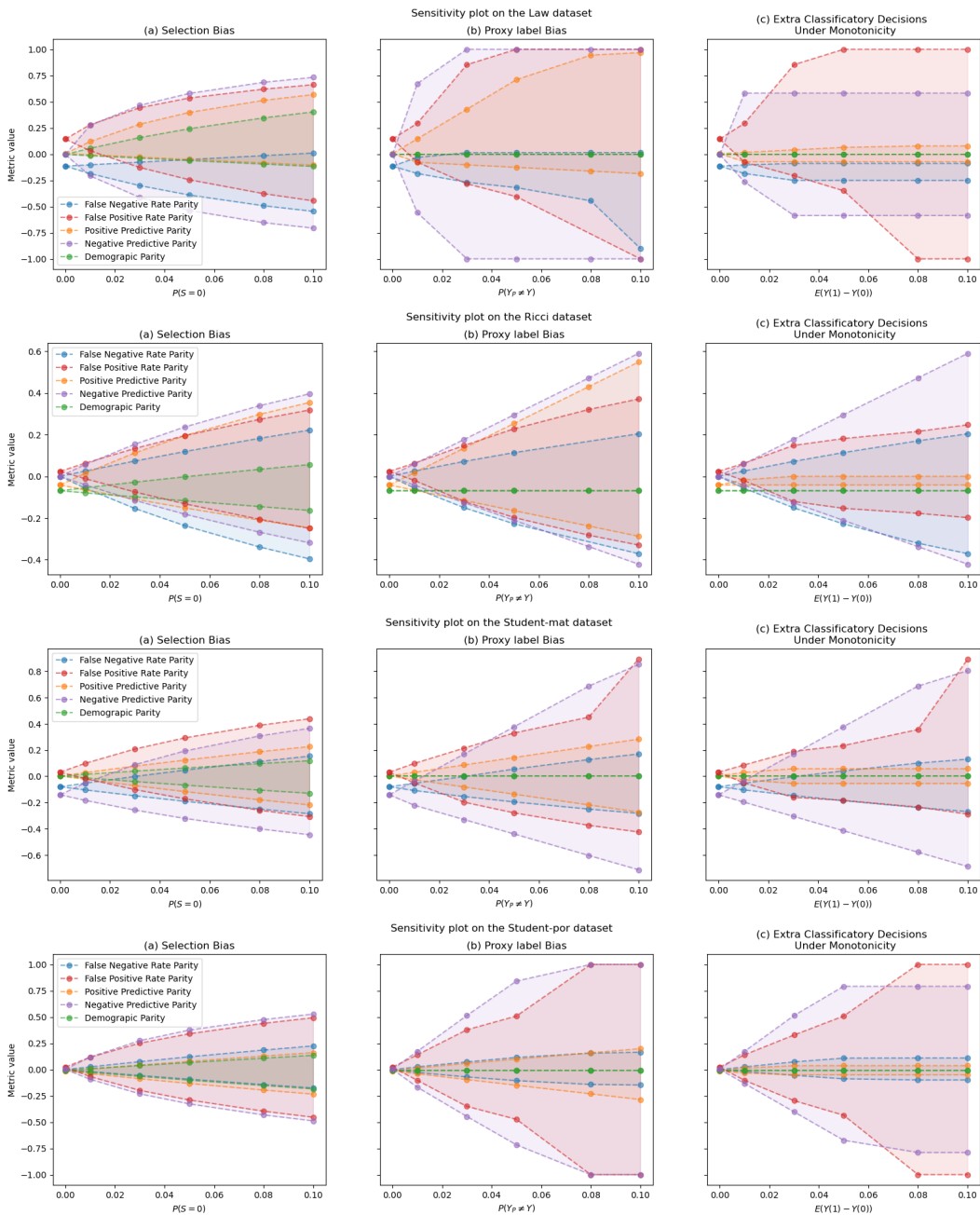

## F.2  Impact Statement

This works aims to broaden the discussion of measurement biases in FairML and provide practical tools for practitioners in the area to use. Our hope is that any potential societal consequences of the work will be positive, corresponding to more equitable algorithmic decision making.