# OpenReview forum: "The Fragility of Fairness: Causal Sensitivity Analysis for Fair Machine Learning"
_NeurIPS.cc/2024/Datasets_and_Benchmarks_Track — NeurIPS 2024 Track Datasets and Benchmarks Poster_

### Official Review · Reviewer_k2PJ · 2024-07-09
**a humbling reminder of the britleness of fairness measures**

**Rating:** 7
**Confidence:** 4

**Review:**

## quality
- paper is not too technical yet formal concepts are used properly
## clarity
- explanations are clear and illustrative examples do help to grasp practical implications
## originality
- even though sensitivity analysis did exist in fairness this work generalizes it to a wider set of biases and metrics
## significance
- this work has the potential to change existing practice in FairML but also in practical Fairness measurement in different domains
- the analogy with the introduction of confidence intervals comes to mind that allows for more robust conclusions (or can be seen as an extension to other sources of randomness)

**Strengths:**

see above

**Additional Feedback:**

I understand that the reduction to the oblivious case can be justified for a wide variety of settings, but I assume not for all. I would be interested in a more thorough discussion on which cases are most consistent with this assumption and which cases - if any - could be problematic.

On extra-classificatory policies I would be interested in knowing more which cases are not covered, ideally with illustrative examples.

**Clarity:**

Real-world examples do a great job in exemplifying discussed notions and anchor the reasoning in readers mind.
Figures legends are too small and difficult to read.

**Correctness:**

- proofs look good to me
- experiments seem legit yet details on hyperparams tuning and other details are a bit lacking

**Documentation:**

- repo code is a bit messy
- otherwise N/A

**Ethics:**

No concern.

**Limitations:**

- runtime of the method in different cases is not discussed
- 3 biases types are described and analyzed yet the coverage of other potential biases is unclear
- repo code is a bit messy

**Opportunities For Improvement:**

- tidy a little bit the repo, there are several unneeded files and incomplete outputs or errors in notebooks
- increase font size in figures legend
- see also questions in Additional Feedback for additional discussion

**Relation To Prior Work:**

Thorough relation to prior work is presented and contextualized for the reader to understand where and how the proposed method improves existing results.

**Summary And Contributions:**

The paper introduces sensitivity analysis to fairness measurement using a causal reasoning framework. This allows to understand how brittle existing work in FairML can be depending on the level of bias in the data, a problem documented in several real world cases. Most decision focused datasets routinely used in FairML works are analyzed and the level of uncertainty due to potential biases is estimated. Results are consistent with existing sensitivity analysis in at least 1 case. An algorithm for sensitivity measurement is provided that works for existing fairness metrics. Implication of this analysis on the robustness of different fairness metrics is discussed.

---

> ### Author Rebuttal · Authors · 2024-08-17
>
> We are deeply appreciative of reviewer **k2PJ**’s affirmation of the significance of the work and their attention to several facets of our work in need of improvements, which we hope to have done justice to in our thorough cleanup of the codebase and our edits to the manuscript for the camera-ready upload.
>
> >tidy a little bit the repo, there are several unneeded files and incomplete outputs or errors in notebooks
>
> >repo code is a bit messy
>
> We refer the reviewer to our general comment where we detail numerous changes to the codebase alongside a webtool to allow practitioners to easily interact with out methodology.
>
> >Figures legends are too small and difficult to read.
>
> >increase font size in figures legend
>
> We have made the legends to all figures and tables of legible size.
>
> >runtime of the method in different cases is not discussed
>
> We have now provided an addendum on runtimes and computational costs. You can get a sense on the website, which has a low-powered CPU and still completes our standard queries in 3-6 seconds per sensitivity parameter value. The most time consuming of our biases is the combined selection and proxy Y, which takes about 1 minute and 30 seconds to run per sensitivity parameter value on an M1 chip.
>
> >3 biases types are described and analyzed yet the coverage of other potential biases is unclear
>
> We have integrated sensitivity analysis for proxy attributes into the codebase and our web interface, as we think that is the most important bias we did not cover.
>
> We also believe both the web interface and the modularity and editability of our configs should make it very easy for practitioners to tailor their analysis to particular biases. Additionally we plan to continue adding bias configs to our repo to develop a real library of known biases, even if we do not have space to discuss them all in the manuscript.
>
> Furthermore we have elaborated our discussion section to point to avenues for future work, including the handling of continous variables going beyond the oblivious case.
>
>
> >I understand that the reduction to the oblivious case can be justified for a wide variety of settings, but I assume not for all. I would be interested in a more thorough discussion on which cases are most consistent with this assumption and which cases - if any - could be problematic.
>
>
> This is an important point. One of the main aims of our paper is to motivate this new line of research and to encourage additional work along these lines, including developing tools to handle cases our framework cannot. Having said this, we provided some suggestions to these ends in the discussions and limitations section. Beyond this, we provide additional references for general causal sensitivity analysis tools which could be used in a similar manner [1,2,3] with [1] specifically focusing on extending the function response framework to continuous variables.
>
> [1] Padh, Kirtan, et al. "Stochastic causal programming for bounding treatment effects." Conference on Causal Learning and Reasoning. PMLR, 2023.
> [2] Frauen, Dennis, Valentyn Melnychuk, and Stefan Feuerriegel. "Sharp bounds for generalized causal sensitivity analysis." Advances in Neural Information Processing Systems 36 (2024).
> [3] Freidling, Tobias, and Qingyuan Zhao. "Optimization-based Sensitivity Analysis for Unmeasured Confounding using Partial Correlations." arXiv e-prints (2022): arXiv-2301.
>
>
> > On extra-classificatory policies I would be interested in knowing more which cases are not covered, ideally with illustrative examples.
>
> The main example which we cannot analyze for ECP is a continuous policy, like if an interest rate were set on a continuous scale as opposed to a discrete low/medium/high. We will make this clearer in the text and add an appendix with this and other examples which lie outside of the scope of our framework.

---

> ### Author Response · Authors · 2024-08-29
> **Prompting reviewer k2PJ for Feedback on our Rebuttal**
>
> We wonder if the reviewer has had a chance to consider our responses, specifically the web tool and clean codebase which address the reviewers limitations on runtime, additional biases and a messy repo.

---

### Official Review · Reviewer_TDXm · 2024-07-24
**Review of "The Fragility of Fairness: Causal Sensitivity Analysis for Fair Machine Learning".**

**Rating:** 6
**Confidence:** 3

**Review:**

Quality:
The paper appears to have a solid background and motivation. The motivation for some more specific choices of the work, such as the marginalization of the features of a dataset are clear and justified.
Clarity:
In terms of clarity, the paper has both highlights and lowlights. While the paper presents in a very clear and understandable manner the possible issues with datasets, the presentation on the causal background and sensitivity analysis might be a bit hard to digest and interpret. Some notation is not completely explained and does not introduce a very relevant concept to the work, in my opinion (e.g., ll. 196-201). Section 4.3 is also a bit unclear, and it does not completely answer the question on how the framework generates results. Figure 1 complements Section 5, and helps understand how we can represent the problems in graph format.
In the results, however, it is not completely clear how the upper bound of the metrics is calculated.
Originality:
The work appears to be original, and a natural derivation of several works previously presented, namely references [13, 19, 24, 32].
Significance of this work:
The work has significant contributions, which may alter the way Fairness is evaluated in Machine Learning.

Pros:
- Relevant and well-motivated topic.
- Adequate explanation of the several issues that dataset may present.
- The conclusions derived from the evaluation have important implications.

Cons:
- Some parts of the paper, especially sections 4 and 6 are not very clear.
- It is not clear how some metrics (upper bound) are obtained.

Additionally, I have to highlight that the repository for this work does not promote reproducibility and extension of the study to other datasets. It is not clear how to adapt the code to perform the same analysis on a specific dataset, nor the methods appear to be standardized to receive different inputs. A readme file with instructions on how to run the experiments and the code would be essential as the objective is for adoption of this framework.

**Strengths:**

The main strengths of the paper appear to be the motivation, significance of the findings, and the approach to solve this issue using causal sensitivity analysis.

**Additional Feedback:**

I believe the presented work has great potential to be used in practice, especially by outputting confidence intervals of the fairness metric for a given interval of the sensitivity parameter. With how the related work was presented, the approach appears to be a natural succession, and future directions are clearly expressed in the manuscript. I believe that some small adjustments in the clarity (especially in sections 4 and 6) would be helpful both for the community of FairML and the adoption by Data Scientists in general. The most important point of improvement in my opinion is the repository. I believe this work fits as a benchmark tool for this track. The tool, however, should be easily accessible and intuitive to run, which is not the case (for now). I think it would also be interesting to add either as supplementary material or in the main body of the paper information regarding computational costs of this approach.

**Clarity:**

The claims made in the submission appear to be correct. The benchmark (dataset and metrics selection) appear to be complete and correct. There should be more information on the models used for predictions in the datasets.

**Correctness:**

The claims made in the submission appear to be correct. The benchmark (dataset and metrics selection) appear to be complete and correct. There should be more information on the models used for predictions in the datasets.

**Documentation:**

The documentation in the repository should be clearer and better organized, with more information on how to reproduce the experiments mentioned in the paper, as well as how to adapt the experiments to a different dataset.

**Ethics:**

There appear to be no ethical concerns regarding this submission.

**Limitations:**

Limitations are clearly pointed out and are aligned with what I understand to be the limitations with this work.

**Opportunities For Improvement:**

The main opportunity for improvement is within the repository and the clarity of the paper. It is not clear how to reproduce experiment results nor is it clear how to adapt the code to a new dataset, which appears to be one of the main objectives of this work.
Additionally, I find it is also important to focus on how to present the results in a clearer manner, and as they are, these are hard to understand and may hinder adoption.

**Relation To Prior Work:**

The work appears to be original, and a natural derivation of several works previously presented, namely references [13, 19, 24, 32].

**Summary And Contributions:**

The authors argue that real world datasets have issues which can render current evaluation practices of fairness faulty. The authors define 3 types of problems that datasets can have, either because a) the measured label does not represent the true label (Proxy bias), b) the selected sample is not representative (Selection bias) or, c) there are exterior decisions that can alter the outcome of an instance (Extra-classificatory Policies). The authors adapt causal sensitivity analysis to the context of these issues and simulate several degrees of severeness on twelve datasets commonly used in FairML. The main conclusions point to the brittleness of the observed fairness metrics under relatively small perturbations to the evaluation sets.
The main contributions of this work can be summarized as a framework for the evaluation of Fairness in Machine Learning, which takes into consideration commonly observed issues in the data. This is made possible by combining the concepts of causal sensitivity analysis and metrics of FairML.

---

> ### Author Rebuttal · Authors · 2024-08-17
>
> We are deeply grateful for reviewer **TDXm**’s appreciation of both the strengths and shortcomings of our manuscript. We hope to have done—and to continue to do—justice to this reviewer’s affirmation that our framework has “great potential to be used in practice.” We address the reviewers main concerns in the following:
>
> >The most important point of improvement in my opinion is the repository. I believe this work fits as a benchmark tool for this track. The tool, however, should be easily accessible and intuitive to run, which is not the case (for now).
>
> >I have to highlight that the repository for this work does not promote reproducibility and extension of the study to other datasets. It is not clear how to adapt the code to perform the same analysis on a specific dataset, nor the methods appear to be standardized to receive different inputs. A readme file with instructions on how to run the experiments and the code would be essential as the objective is for adoption of this framework.
>
> >The main opportunity for improvement is within the repository and the clarity of the paper. It is not clear how to reproduce experiment results nor is it clear how to adapt the code to a new dataset, which appears to be one of the main objectives of this work.
>
> >The documentation in the repository should be clearer and better organized, with more information on how to reproduce the experiments mentioned in the paper, as well as how to adapt the experiments to a different dataset.
>
> We thank the reviewer for these important points. In response to these we have made the following substantantial changes which we discuss in the general comment and reproduce here for the convinience of the reviewer:
>
> 1) We have greatly cleaned our codebase, added clear documentation and easy to use notebooks allowing for the recreation of all experiments in the paper. Our new repo can be found [here](https://anonymous.4open.science/r/fragility_fairness).
> 2) We have added a detailed readme and a notebook demonstrating how to use the codebase.
> 3) We have created an accessible web interface, allowing for our toolkit to be used by practitioners without having to interact directly with the codebase. This web tool allows practitioners to upload their own data, ensuring our methodology can be applied on custom datasets.
>
> This is also the focus of our general comment above.
>
> > The presentation on the causal background and sensitivity analysis might be a bit hard to digest and interpret.
>
> >Some parts of the paper, especially sections 4 and 6 are not very clear.
>
> >Section 4.3 is also a bit unclear, and it does not completely answer the question on how the framework generates results.
>
> >It is not clear how some metrics (upper bound) are obtained.
>
> We thank the reviewer for raising this point. As mentioned in the general comment, we have rewritten the technical section (4) of the paper to ensure that it is more approachable to practitioners without a background in causal inference. We have moved technical definitions of SCM and marginalisaion to the appendix, and replaced them with DAGs explaining each of the concepts. This ensures practitioners can understand the key details without getting bogged down in overly technical notation.
>
> We have also made the computation of the bounds more transparent. The autobounds framework calculates bounds via a branch and bound procedure, using standard methods for bounding rational polynomials over compact spaces. Branch and bound optimization procedures simultaneously calculate lower and upper bounds for the target, and terminate the optimization when these are equal. This ensures that when solving the optimization we have valid upper bounds and so produce valid bounds on any query at a given step. We use Couenne as the back-end algorithm to implement the branch and bound optimization. This can be slow, and in practice we find we can use generic non-linear optimizers like ipopt to get identical results. For the main experiments we certify the results via Couenne, but the primary optimizer we suggest for most analyses is ipopt.
>
> >I think it would also be interesting to add ... computational costs of this approach.
>
> We have now provided an addendum on runtimes and computational costs. You can get a sense on the website, which has a low-powered CPU and still completes our standard queries in a matter of seconds per sensitivity parameter value. The most time consuming of our biases is the combined selection and proxy Y, which takes about 1 minute and 30 seconds to run per sensitivity parameter value on an M1 chip.
>
> > There should be more information on the models used for predictions in the datasets.
>
> Model details for the experiments can be found in Appendix F of our supplementary material, but we will clarify this in the main text.
>
> > I find it is also important to focus on how to present the results in a clearer manner...
>
> If accepted, we will use our extra page for the presentation of the experimental section, with a clearer analysis of results.
>
> >Some notation is not completely explained
>
> Thank you for noting this. We have changed the paper to ensure all notation is included in the notation section or given a definition before used.

---

> ### Author Response · Authors · 2024-08-29
> **Prompting reviewer TDXm for Feedback on our Rebuttal**
>
> We wondered if the reviewer has had an opportunity to view our responses and if this changed their assessment of our paper? We would like to draw specific attention to the accessible web interface and extensive cleaning of our codebase which were targeted at the concerns regarding reproducibility and barriers to adoption.

---

> > ### Comment · Reviewer_TDXm · 2024-08-29
> >
> > I did not have access to the revised version of the manuscript, but the authors seem willing to address my main concerns.

---

> > > ### Author Response · Authors · 2024-08-29
> > >
> > > We thank the reviewer for acknowledging our rebuttal. In light of the fact that we have addressed the reviewers concerns and that the reviewer recognises that our paper has significant contributions, we hope the reviewer would agree the paper merits acceptance. If so, we wonder if the reviewer would consider increasing their score since the current score is below the acceptance threshold.

---

### Official Review · Reviewer_5n81 · 2024-07-24
**Causal Sensitivity Analysis Reveals Measurement Biases in FairML Datasets**

**Rating:** 7
**Confidence:** 4

**Review:**

The paper applies recent causal sensitivity analysis tools to analyze measurement biases in FairML datasets. It considers three types of measurement biases: proxy label bias, selection bias, and extra-classificatory policies. The paper finds that all 13 considered datasets suffered from one or another form of measurement bias.

The sensitivity analysis conducted in this study is of immense practical value. It sheds light on the constraints of current FairML datasets and underscores the importance of being aware of and accounting for biases in real-world FairML applications.

**Strengths:**

- I enjoyed reading the paper. It introduced complex fairness and causal inference concepts through clear and lucid examples.

- The application of causal sensitivity analysis to FairML since it has not been traditionally applied to FairML problems.

- The experiments, spanning a variety of datasets and fairness metrics, lead to valuable and inspiring conclusions for designing new fairness benchmarks and for real-world applications of FairML.

**Additional Feedback:**

- Clearly highlighting the contributions of the paper would make the paper stronger.
- Consider performing the sensitivity analysis on the Folktables[1] dataset, which is the current standard benchmark dataset for Algorithmic Fairness.

[1] Retiring Adult: New Datasets for Fair Machine Learning, NeurIPS 2021

**Clarity:**

The paper is well-written and exceptionally clear. The lucid examples make complex concepts in fairness and causal inference easily understandable.

**Correctness:**

The claims and application of the causal sensitivity analysis on FairML datasets appear sound and correct. The

**Documentation:**

No dataset proposed in the paper.

**Ethics:**

No ethics concerns.

**Limitations:**

- **Contributions:** The scope of the paper is limited to identifying the extent to which these biases affect fairness metrics. The paper does not provide a solution or suggestions. This, per se, is not a weakness, but in the context of existing work, this paper only seems like a comprehensive view of prior work that also does not provide any solution.

- **Generality:** As mentioned in lines 268-270, the causal graphs for each problem will vary in the context, and therefore, modeling each bias would vary. One may look at this as both a positive and a negative quality. As a positive quality, this means the method is generic enough to be applied in various cases. As a negative quality, this means subjective knowledge of the downstream problem/domain/task (in the form of a causal graph) is required to conduct the sensitivity analysis. Very often, the causal graph may not be fully or reliably known.

**Opportunities For Improvement:**

- Clearly highlighting the contributions of the paper would make the paper stronger.
- Consider performing the sensitivity analysis on the Folktables[1] dataset, which is the current standard benchmark dataset for Algorithmic Fairness.

[1] Retiring Adult: New Datasets for Fair Machine Learning, NeurIPS 2021

**Relation To Prior Work:**

While the causal sensitivity tools used in the paper are clearly discussed, the discussion of the closest related work and contributions of the current paper is missing.

**Summary And Contributions:**

The dataset collected from the real world will have measurement biases (of which the paper considers three). These biases could influence the parity metrics that are generally used to measure fairness. The extent to which these parity metrics are affected by such biases is studied as a causal sensitivity analysis problem. For sensitivity analysis, the bias is used as the parameter. The proposed method provides upper and lower bounds for the fairness metric.

---

> ### Author Rebuttal · Authors · 2024-08-17
>
> We are deeply appreciative of reviewer **5n81**’s positive assessment of our work, and are glad to hear that they enjoyed reading the work. We provide the following which we hope will resolve the reviewers remaining concerns and improve the paper:
>
>
> >Clearly highlighting the contributions of the paper would make the paper stronger.
>
> We view the key contributions of the paper as follows:
> 1. Demonstrating that the evaluation of fairness metrics often falls victim to a number of common fairness metrics. We show in our cross dataset analysis that at least one bias we reference is present in each of the standard FairML datasets and 60% of the time all three are present. Alongside this we pull out some of the nuances involved in defining an "unbiased" population in each of these cases.
> 2. In response to this we developing a practical framework, based in causal sensitivity analysis, which allows practitioners to understand the effect that such biases would have on the evaluation of fairness metrics.
> 3. We experimentally validate our framework by performing empirical analyses on benchmark datasets in the fairness literature which reveals the sensitivity, or “fragility” of fairness metrics to these biases.
>
> >Consider performing the sensitivity analysis on the Folktables[1] dataset, which is the current standard benchmark dataset for Algorithmic Fairness.
>
> >[1] Retiring Adult: New Datasets for Fair Machine Learning, NeurIPS 2021
>
> We are very appreciative of the suggestion! We have now run our analyses on Folktables for a total of 14 datasets analysed.
>
> >The scope of the paper is limited to identifying the extent to which these biases affect fairness metrics. The paper does not provide a solution or suggestions. This, per se, is not a weakness, but in the context of existing work, this paper only seems like a comprehensive view of prior work that also does not provide any solution.
>
> The core takeaway is that the results of fairness evaluations, absent further contextual information, mean very little on their own.
>
> We can do better than this though: our tool is much more flexible than prior work, so practitioners can develop a clear understanding of exactly which biases or assumptions are important to their analyses being stable/robust. On its own this may be a solution in many cases, allowing practitioners to pinpoint the assumptions under which their analyses hold. If those assumptions are still too strong our tool can help practitioenrs determine how to collect better data so their analyses can be improved. We have added a more detailed discussion of these points to the manuscript.
>
>
> >Generality: As mentioned in lines 268-270, the causal graphs for each problem will vary in the context, and therefore, modeling each bias would vary. One may look at this as both a positive and a negative quality. As a positive quality, this means the method is generic enough to be applied in various cases. As a negative quality, this means subjective knowledge of the downstream problem/domain/task (in the form of a causal graph) is required to conduct the sensitivity analysis. Very often, the causal graph may not be fully or reliably known.
>
> This is perhaps the most important point of our work: assumptions about the data generating process are necessary in order for fairness evaluations to have any meaning relative to what the practitioner hopes to measure. Therefore we view the flexability as a strength of our framework as it allows practitioners to vary assumptions according to domain specific knowledge.
>
> That being said, we do not need a fully specified causal graph to make progress. A general enough causal graph can be chosen such that no particular restrictions are implied if the bias and statistic are purely observational. This can be seen in our recreation of the Fogliato et al paper, where the original paper does not use any causal graphs. In this case our general proxy graph does not imply any additional constraints and so we can recover their results. We will make this link clearer in a revised version of the paper.
>
> Finally if there are several DAGs which seem plausible it is straightforward to run the analysis under each possilbe DAG. If the results are similar then the answer is clear. If results differ significantly then the analysis will point to key aspects of the data generating process which analysts should aim to understand in order to know if their models are reliable.

---

> > ### Comment · Reviewer_5n81 · 2024-08-21
> >
> > The rebuttal has addressed the concerns raised in the initial review. I have increased my score.
> >
> > I hope the authors include the results on Folktables and a clear description of the contributions in the revised paper.

---

### Official Review · Reviewer_smoh · 2024-07-25
**Review of "The Fragility of Fairness: Causal Sensitivity Analysis for Fair Machine Learning"**

**Rating:** 8
**Confidence:** 2

**Review:**

Quality: the paper is of high quality, presenting a well-structured framework and thorough analysis. The experiments are well-designed, and the results are well discussed, showing the practical implications of the proposed framework. However, the complexity of the methodology might pose challenges for practitioners without a strong background in causal inference.

Clarity: the paper is generally well-written, but some sections, particularly those explaining technical details, could benefit from additional simplification or illustrative examples. The authors should also define symbols when they are first introduced to improve clarity.

Originality: the approach of using causal sensitivity analysis in the context of fair ML is somewhat new. The framework provides a new perspective on evaluating fairness metrics, highlighting the importance of accounting for measurement biases and how fragile fairness metrics can be.

Significance: the findings have significant implications for the field of fair ML, demonstrating that fairness metrics can be highly sensitive to even small levels of bias. This work encourages more robust and nuanced evaluations of fairness in ML models, making it highly relevant to the broader research community.

**Strengths:**

* The proposed general framework provides a new and critical perspective on evaluating fairness in ML, addressing a crucial aspect of fairness metric sensitivity that is often overlooked.
* The findings and the proposed methodology are highly relevant to researchers and practitioners in ML, particularly those working on fairness and bias mitigation. Any new proposal for fairness metrics should consider evaluating its sensitivity with a framework of this kind.
* By highlighting the fragility of fairness metrics and the importance of accounting for biases, the paper contributes to the development of more equitable and reliable ML.

**Additional Feedback:**

N/A

**Clarity:**

The paper is well-written overall, but there are some typos and some sections could be improved for clarity. Defining symbols when first introduced and simplifying technical explanations would enhance readability (first time (A, Y, ^Y) is mentioned)
The paper mentions 12 canonical datasets in the abstract and Section 6.3 but 13 benchmark datasets in Table 1 and Section 3.4. This should be clarified to avoid confusion.

Some typos:
Line 57: "one the problems" should be "on the problems".
Line 76: "how to asses" should be "how to assess".
Line 89: "which their exists" should be "which there exists".
Line 180: "where we no not" should be "where we do not".
Caption of Figure 2: "derived bounds recover the algebraically" should be "derived bounds to recover the algebraically".

**Correctness:**

The claims made in the submission seem correct, to the best of my knowledge. The methodology seems sound, and the experiments are appropriately designed and executed. The conclusions drawn from the analyses seem valid and grounded by the results.

**Documentation:**

The supplementary material provides sufficient detail on the used datasets and corresponding results. There's an available repository, Ibut documentation is not very thorough.

**Ethics:**

There are no significant ethical concerns with this paper. The authors are transparent about the used datatsets and provide a repository for reproducibility. Their work aims to improve the ethical and social implications of ML, so it is reasonable to assume the potential societal impacts of the work are positive, even though a more clear discussion could benefit the paper.

**Limitations:**

The authors have acknowledged some of the main limitations of this work. They acknowledge the assumptions and constraints of their framework and emphasize the importance of accounting for measurement biases. However, further discussion on extending the framework beyond the current constraints would be beneficial. Although it is somewhat easy to see the positive potential of the framework, it would be important for the authors to discuss what potential negative societal impacts (even if minor) their work might entail. For instance, if there is lack of clarity or guidance in the toolkit, the practitioner could incur in wrong conclusions about the results of the studied fairness metrics, leading to a false sense of "security".

**Opportunities For Improvement:**

* The framework is currently limited to discrete variables and the oblivious setting. Providing hints or guidance on extending the framework beyond these constraints would enhance its applicability.
* Simplifying some of the technical details and providing more illustrative examples would make the methodology more accessible to a broader audience.
* Including other types of biases in the analysis could provide a more extensive understanding of ML fairness.
* The authors should clarify the number of datasets analyzed (12 vs. 13) to avoid confusion.
* The paper needs a revision, as there are numerous typos in the text.

**Relation To Prior Work:**

The paper provides a comprehensive review of related work and clearly positions the current study within the context of existing research, highlighting the novelty and relationships with previous contributions.

**Summary And Contributions:**

This work introduces a framework for methods adapted from causal sensitivity analysis to evaluate the robustness of fairness metrics in machine learning models. The authors focus on three main types of biases: proxy (label) bias, selection bias, and bias from extra-classificatory policies (ECPs). The analysis allows any combination of fairness metric an bias which can be posed in the "oblivious setting", where the covariates are unknown. They demonstrate that even minor biases can significantly impact fairness assessments, making many fairness metrics fragile. The framework is applied to 3 different types of classifiers and various datasets, revealing the necessity for more robust fairness evaluations in machine learning.

The main contributions are:

* Development of a (graphical) causal sensitivity analysis framework adaptable to various fairness metrics and biases within the "oblivious setting".
* Analysis of the sensitivity of common fairness metrics under different classifiers across multiple known fairness-related datasets.
* Demonstration of the significant impact of measurement biases on the robustness of fairness evaluations.
* A toolkit for ML practitioners and researchers to evaluate the implications of measurement biases on fairness metrics (available repository).

---

> ### Author Rebuttal · Authors · 2024-08-17
>
> We are deeply appreciative of reviewer **smoh**’s thorough and positive review of our paper. We believe that the manuscript and codebase have been much improved in light of this reviewer’s detailed feedback. Response in what follows.
>
>
> >The paper is generally well-written... additional simplification or illustrative examples.
>
> >The complexity of the methodology...
>
> >Simplifying some of the technical details...
>
>
> We thank the reviewer for raising this point. We would like to refer to the general comment where we have made multiple changes, focused around improving readability and making our work and code accesible to less technical readers. Broadly these changes are:
>
> 1. A major refactoring of the code base to allow figures for significantly improved reproducability. We also provide notebooks to reproduce each figure in the text.
> 2. Creating an interactive webtool which allows practitioners to easily interact with the tools we develop. This lets practitioners use their own data, express their own biases and add their own constraints whilst being fully interoperable with the rest of our code.
> 3. Rewriting section 4, moving the most technical definitions to appendicies and instead explaining the definitions using causal graphs.
>
>
> >Including other types of biases in the analysis could provide a more extensive understanding of ML fairness.
>
> We took this point very seriously in our response to this review. We have integrated sensitivity analysis for proxy attributes into the codebase and our web interface, as we think that is the most important bias we did not cover.
>
> We also believe both the web interface and the modularity and editability of our configs should make it very easy for practitioners to tailor their analysis to particular biases. Additionally we plan to continue adding bias configs to our repo to develop a real library of known biases, even if we do not have space to discuss them all in the manuscript.
>
> Furthermore we have elaborated our discussion section to point to avenues for future work, including the handling of continous variables going beyond the oblivious case.
>
> >Further discussion on extending the framework...
>
> >The framework is currently limited to discrete variables and the oblivious setting...
>
>
> This is an important point. One of the main aims of our paper is to motivate this new line of research and to encourage additional work along these lines, including developing tools to handle cases our framework cannot. Having said this, we provided some suggestions to these ends in the discussions and limitations section. Beyond this, we provide additional references for general causal sensitivity analysis tools which could be used in a similar manner [1,2,3] with [1] specifically focusing on extending the function response framework to continuous variables.
>
> [1] Padh, Kirtan, et al. "Stochastic causal programming for bounding treatment effects." Conference on Causal Learning and Reasoning. PMLR, 2023.
> [2] Frauen, Dennis, Valentyn Melnychuk, and Stefan Feuerriegel. "Sharp bounds for generalized causal sensitivity analysis." Advances in Neural Information Processing Systems 36 (2024).
> [3] Freidling, Tobias, and Qingyuan Zhao. "Optimization-based Sensitivity Analysis for Unmeasured Confounding using Partial Correlations." arXiv e-prints (2022): arXiv-2301.
>
>
> > [The authors should discuss] potential negative societal impacts... lack of clarity [could lead] to a false sense of "security".
>
> One of our primary aims in this piece, in addition to providing our testing toolkit, is to engender awareness of inherent limitations on the informativeness and utility of the fairness framework, or any simplistic, quantitative approach to operationalising ethical frameworks in general. That being said, we agree that if this tool became standard in the field it might run the risk of becoming, like parity metrics themselves, a kind of "fairwashing" [4] that leads to a false sense of security as the reviewer suggests. We think this is clearly outweighed by the positive impacts of addressing systematic databiases, and we hope the contextual nature of the tool will ameliorate these worries, but we have amended our impact statement to include this concern.
>
> [4] Aïvodji, Ulrich, et al. "Fairwashing: the risk of rationalization." International Conference on Machine Learning. PMLR, 2019.
>
> >The authors should clarify the number of datasets analyzed (12 vs. 13) to avoid confusion.
>
> We have now analysed a total of 14 benchmark datasets (13 originally and the additional Folktables dataset as suggsted by reviewer 5n81) and reported this consistently throughout the text.
>
>
> >The authors should also define symbols...
>
> We have made certain in revisions to introduce all key concepts, variables, and notational conventions as they appear in the text.
>
> >The paper needs a revision...
>
> We have carefully gone through the manuscript to correct grammar and spelling throught, apologies for this issue!

---

> > ### Comment · Reviewer_smoh · 2024-08-29
> >
> > I thank the authors for a thorough response, and for the major effort in improving the codebase, and building an interactive webtool (I cannot access due to security reasons related to ssl certificate). I cannot read the revised paper but appreciate the effort in making the paper more accessible to a broader audience by simplifying some of the technical definitions, and using causal graphs.

---

> > > ### Author Response · Authors · 2024-08-29
> > > **Response to Reviewer smoh's comment**
> > >
> > > We thank Reviewer smoh for their comment, the web tool should be accessible and we find it accessible in all of our testing. If the reviewer could clarify the issue we would love to correct it.
> > >
> > > Additionally if we have resolved core weaknesses in our response we would ask if the reviewer might consider raising their score.

---

> > > > ### Comment · Reviewer_smoh · 2024-09-02
> > > >
> > > > I think the issue was the lack of an ssl certificate, but now I could access the interface. I've also raised the score accordingly.

---

### Author Rebuttal · Authors · 2024-08-21

We would like to thank all of our reviewers for their time and for their positive and constructive feedback on our paper. The work was well received, with every reviewer highlighting, in particular, its potential impact:

- **Reviewer TDXm**: "The work has significant contributions, which may alter the way Fairness is evaluated in Machine Learning."

- **Reviewer k2PJ** : "This work has the potential to change existing practice in FairML but also in practical Fairness measurement in different domains".

- **Reviewer smoh**:  "The findings have significant implications for the field of fair ML, demonstrating that fairness metrics can be highly sensitive to even small levels of bias. This work encourages more robust and nuanced evaluations of fairness in ML models, making it highly relevant to the broader research community."

- **Reviewer 5n81** : "I enjoyed reading the paper...The experiments, spanning a variety of datasets and fairness metrics, lead to valuable and inspiring conclusions for designing new fairness benchmarks and for real-world applications of FairML."


Reviewers consistently pointed to two shortcomings that might hold the work back: (1) Our codebase was not well documented nor very usable at time of submission, and (2) Parts of our manuscript were not maximally approachable for less-technical readers. Here we detail chages we have made to resolve these issues as follows:

1) We have completely refactored our codebase to allow for portability, modularity, and reproducibility. Our new repo can be found [here](https://anonymous.4open.science/r/fragility_fairness). We will release a python package for this improved codebase to make installation easy. Specific changes we have made:
    - We have developed a way to represent biases as json configs, which are easy to understand and use, hugely lowering the barrier to users developing and sharing their own biases.
    - We have written two core interfacing functions which can take bias configs and metrics and compute sensitivity on a given dataset.
    - We have rewritten all of our experiments to use this interface based on a unified set of configs representing our biases, and provided self-contained Jupyter Notebooks which can recreate each individual figure from the manuscript.
    - We have added a readme which explains our configs and the core functions of the repo, as well as documentation elsewhere throughout the repo.
2) We have created an accessible web interface, allowing for our toolkit to be used by practitioners without having to interact directly with the codebase at all: [fragile.ml](https://fragile.ml). This web tool:
    - Has a full user interface where practitioners can develop their own biases and explore assumptions in an intuitive way.
    - Allows practitioners to upload their own datasets and create sensitivity plots using their own biases and datasets.
    - Can export and import bias configs which are fully interoperable with the rest of the codebase.

We believe these changes not only ensure practitioners can easily check the sensitivity of our particular biases but, in the spirit of our work, also allows non-technical practitioners to develop biases accounting for their particular data and assumptions. We hope these changes address the reviewers concerns and provide a useful tool for the wider Fair-ML community.

Finally, we have rewritten the technical section (4) of the paper  to ensure that it is more approachable to practitioners without a background in causal inference. We have moved technical definitions of SCM and marginalization to the appendix, and replaced them with DAGs giving pictoral explanations of each of these concepts. This ensures practitioners can understand the key details necessary to use our toolkit without getting bogged down in overly technical notation.

---

### Decision · Program_Chairs · 2024-09-26

**Decision:**

Accept (Poster)

**Comment:**

This paper introduces a framework adapted from causal sensitivity analysis to evaluate the robustness of fairness metrics. It is well-written and includes a comprehensive analysis. The topic has the potential for broad impact. Thus, I recommend acceptance. Please ensure that the responses to reviewers during rebuttal are included in the revised version.